# 𝒳Transplant: A Probe into the Upper Bound Performance of Multilingual Capability in LLMs via Cross-lingual Transplantation

## Abstract

Current large language models (LLMs) often display significant imbalances in their multilingual capabilities and cultural adaptability, primarily due to their unbalanced and English-centric pretraining data. For these English-centric LLMs, the disparities between English and non-English languages hinder their ability to utilize their robust English-based capabilities within non-English contexts, while also limiting access to valuable multilingual knowledge derived from non-English "language-specific neurons" within English contexts. Motivated by this, our work explores the possibility for LLMs to leverage the strengths of both English and non-English languages, aiming to further unlock their multilingual potential. To this end, we propose a probing method named 𝒳Transplant, which directly transplants feed-forward activations from English input to non-English (or from non-English to English) during inference stage, allowing the model to benefit from both English and additional multilingual knowledge. Through extensive experiments on our pilotsets and representative LLMs across different tasks and languages, we empirically prove that both the multilingual capabilities and cultural adaptability of LLMs hold the potential to be significantly improved by the cross-lingual feed forward transplantation, respectively from En → non-En and non-En → En. Additionally, we also establish the upper bound performance of LLMs obtained through 𝒳Transplant (relative growth of **+80%** in multilingual capabilities, **+39%** in cultural adaptability), highlighting the underutilization of current LLMs' multilingual potential. We do hope our further analysis and discussion could suggest promising directions for deeply unlocking the multilingual potential of current English-centric LLMs.

## 1 Introduction

In recent years, large language models (LLMs) have showcased their remarkable versatility across a wide range of downstream tasks (Zhao et al., 2023; Liu et al., 2023; Dong et al., 2023; Wei et al., 2022a;b; Shanahan, 2022), as well as their evident generalizability and adaptability in multilingual scenarios. However, the significant imbalances in their multilingual capabilities and cultural adaptability still remain challenges that researchers are striving to resolve (Ye et al., 2023; Li et al., 2024a; Shi et al., 2024; Qin et al., 2024). These issues primarily stem from their unbalanced training corpora, which is predominantly in English, leading to these models being termed *English-centric* LLMs (Brown et al., 2020; Zhang et al., 2022; Touvron et al., 2023; Biderman et al., 2023).

Existing methods for these challenges primarily focus on *Multilingual Pretraining* and *Cross-lingual Transfer*. Multilingual Pretraining involves initially or continuously training models on diverse multilingual datasets to develop an overall improvement of their multilingual capabilities (Lin et al., 2021; Scao et al., 2022; Gao et al., 2024; Li et al., 2024b). While Cross-lingual Transfer leverages knowledge from high-resource languages to enhance the performance of low-resource languages through fine-tuning techniques (Reid & Artetxe, 2022; Cahyawijaya et al., 2023; Ye et al., 2023; Khurana et al., 2024). However, these training-based methods have shown potential limitations. Conneau et al. (2020) identified the "curse of multilinguality", a form of negative interference (Wang et al., 2020), where expanding too much languages during pretraining eventually leads to a decline.

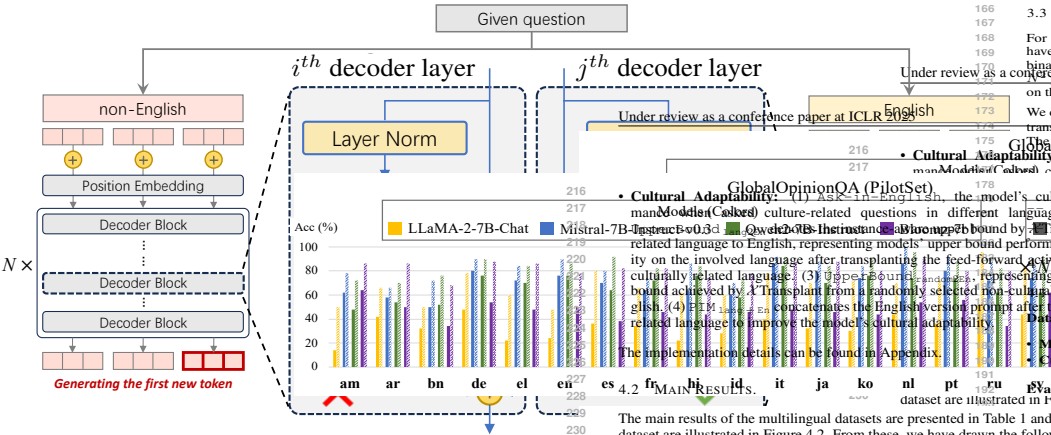

Figure 1: Overview of the mechanism of $\mathcal{X}$Transplant, **taking the direction of En $\rightarrow$ non-En as the example**. During the prediction of the first token with a non-English prompt, the feed-forward activations from a specific decoder layer in English are leveraged to replace the original one in a layer of the non-English input, with the forward propagation to proceed with the transplanted activations.

Given these limitations, current methods still struggle to effectively and efficiently address the pronounced imbalances in multilingual capabilities and cultural adaptability of LLMs. This situation also places humans in a dilemma with current English-centric LLMs: given a certain question, (1) posing in English may overlook the language-specific neurons that is only activated by non-English inputs, potentially resulting in incomplete or inaccurate responses. On the other hand, (2) posing in non-English languages may fail to leverage the model's strong general capabilities in English, thereby affecting its overall performance. This naturally leads to a key consideration:

> *Can the LLMs leverage both their powerful general capabilities (in English) and their (non-English) multilingual knowledge during inference, to fully unlock their multilingual potential?*

In response to this, we first introduce and investigate a probing method named $\mathcal{X}$Transplant, which diverges from traditional training-based approaches, to explore this possibility through cross-lingual transplantation. As shown in Figure 1, during the prediction of the first new token, $\mathcal{X}$Transplant transplants the feed-forward activations of certain decoder layer in English into the inference process of non-English input (or from non-English into English), with the forward propagation and subsequent token generation to proceed with the transplanted activations. The goal is to enable the model to benefit from English capabilities during non-English inference (and vice versa, allowing English to benefit from non-English knowledge). By leveraging this probe, our study delves into two distinct avenues: the impact of En $\rightarrow$ non-En transplantation on LLMs' multilingual capabilities, and how non-En $\rightarrow$ En transplantation affects LLMs' non-English cultural adaptability.

We conduct extensive experiments and analysis on four representative LLMs across the pilotsets of three multilingual datasets and one culture-aware dataset. By assessing the upper bound performance of LLMs obtained through exhaustively evaluating all possible settings of $\mathcal{X}$Transplant, we empirically demonstrate that $\mathcal{X}$Transplant hold the potential to push the boundaries of what LLMs can typically achieve in multilingual and culture-aware tasks, with an average upper bound 30% higher than the original, which highlights the underutilization of current LLMs' multilingual potential. We also undertake some analysis and discussion concerning further unlocking LLMs' multilingual potential, with the hope of providing insights for future research.

## 2 BACKGROUND AND HYPOTHESIS

In this section, we provide a detailed overview of the background that motivates our research and clearly articulate the central hypothesis underpinning our proposed probing approach.

**Decoder-only Large Language Models** The transformer-based GPT series of models have shown remarkable effectiveness in natural language generation (Radford et al., 2018; Brown et al., 2020), triggering a boom around LLMs. Within Transformer (Vaswani et al., 2017), the feed-forward

layers and self-attention module constitute the main body of a decoder block for current LLMs. Let $x$ denote the input matrix, a decoder-only layer can be mainly formulated as follows:

$$\text{AttRes}(x) = \text{Concat}(\text{Self-Att-head}_1(x), \ldots, \text{Self-Att-head}_\text{h}(x))W_O + x \quad (1)$$

$$\text{DecoderBlock}(x) = \text{FFN}(\text{AttRes}(x)) + \text{AttRes}(x) \quad (2)$$

where $\text{Self-Att-head}_\text{h}(x)$ represents a single attention head, $\text{Concat}$ concatenates all heads followed by a projection with $W_O$, and $\text{AttRes}(x)$ is the intermediate state obtained by adding a residual connection to the projected heads. The feed-forward module is denoted by $\text{FFN}$. The whole decoder output, is obtained by applying a residual connection between $\text{AttRes}(x)$ and the output of FFN.

The background, that the feed-forward layers have been shown in many studies to play a crucial role in storing factual knowledge (Geva et al., 2021; Dai et al., 2022; Meng et al., 2022), provides the reason for why we explore cross-lingual transplantation on **feed-forward layers**, which aligns with our goal of enabling LLMs to fully leverage both English and non-English multilingual knowledge.

**Language-specific Neurons** The intriguing capability of LLMs to understand and generate text in various languages is attributed to a subset of neurons within their architecture that exhibit high activity levels when processing specific languages. Termed as "language-specific neurons", these components have been identified as pivotal to the multilingual competencies of LLMs (Tang et al., 2024; Kojima et al., 2024). Furthermore, the proportion of these neurons is notably small relative to the entire neural network, yet their targeted activation or deactivation significantly impacts the model's ability to understand and generate language-specific content (Zhao et al., 2024). This finding has profound implications for enhancing LLMs' multilingual capabilities.

Building on the theoretical foundations regarding feed-forward layers and language-specific neurons, we boldly hypothesize that sharing and transferring feed-forward activations between English and non-English languages may allow the model to leverage the strengths of both language groups. This capacity to integrate advantages from diverse linguistic backgrounds serves as the foundation of our probing method—$\mathcal{X}$Transplant.

# 3 $\mathcal{X}$TRANSPLANT: CROSS-LINGUAL TRANSPLANTATION

In this section, we will present the formulation of $\mathcal{X}$Transplant, elaborate on its implementation details, and delineate several relevant concepts.

## 3.1 METHODOLOGY

For a model $M$ with $N$ decoder layers, given an original input $x_s$ in source language $S$, the $x_s$ undergoes a forward propagation through all decoder layers to predict the next token. Let the output activations of these $N$ decoders be denoted as $O_s = \{o_s^k\}_{k=1}^N$, where each $o_s^k$ is obtained by combining the feed-forward activations $f_s^k$ and self-attention activations $a_s^k$ through a residual connection (as shown in Equation 2). Similarly, for another version of $x_s$ in target language $T$, denoted as $x_t$, we also have $O_t = \{o_t^k\}_{k=1}^N$ with corresponding $\{f_t^k\}_{k=1}^N$ and $\{a_t^k\}_{k=1}^N$. Without any modifications, they would predict the first new token $\hat{y}_s$ and $\hat{y}_t$ with the unembed matrix $W_{unembed}$ as follows:

$$\hat{y}_s = \text{softmax}(W_{unembed} \cdot o_s^N) = \text{softmax}(W_{unembed} \cdot (a_s^N + f_s^N)) \quad (3)$$

$$\hat{y}_t = \text{softmax}(W_{unembed} \cdot o_t^N) = \text{softmax}(W_{unembed} \cdot (a_t^N + f_t^N)) \quad (4)$$

Our approach, $\mathcal{X}$Transplant, refines the process by transplanting the feed-forward activations from the $i^{th}$ decoder layer with input $x_s$ to the $j^{th}$ decoder layer with input $x_t$. Formally, $f_t^j$ is replaced with $f_s^i$ and the forward propagation of prompting $x_t$ then continues with this modification. As a result of this operation, the original $\{o_t^k\}_{k=j}^N$ will be altered into $\{\tilde{o}_t^k\}_{k=j}^N$ due to the update in $f_t^j$, leading to new prediction outcomes $\hat{y}_t^{(\text{modified})}$ as follows:

$$\hat{y}_t^{(\text{modified})} = \text{softmax}(W_{unembed} \cdot \tilde{o}_t^N) \quad (5)$$

Notably, $\mathcal{X}$Transplant currently considers only the substitution of feed-forward activations from a single layer, meaning that the aforementioned $i^{th}$ layer and $j^{th}$ layer both refer to a certain, single decoder layer. Besides, $\mathcal{X}$Transplant performs the transplantation only during the forward propagation for predicting the first new token; all subsequent tokens are generated iteratively after the first one, without any additional transplantation operations.

## 3.2 Mutual Transplantation

Section 3.1 details how $\mathcal{X}$Transplant facilitates the transfer of feed forward activations from language $S$ to language $T$. But $\mathcal{X}$Transplant actually supports transplantation in two directions. When prompting in non-English, the feed-forward activations from English can be leveraged to help the process of non-English prompting. Similarly, under the English prompting conditions, the feed-forward activations from non-English languages can be leveraged to help. Specifically, our experiments explore the dual attempt of $\mathcal{X}$Transplant: $\texttt{En} \rightarrow \texttt{non-En}$ and $\texttt{non-En} \rightarrow \texttt{En}$.

## 3.3 Instance-ware Upper Bound

For a model $M$ with $N$ decoder layers, both the source layer and target layer selections in $\mathcal{X}$Transplant offer $N$ possible choices, resulting in $N^2$ potential transplantation combinations. For a dataset $D$ of a certain size, we conducted $\mathcal{X}$Transplant for each sample across all $N^2$ possibilities, selecting the optimal solution for each instance. The model's optimal performance on this dataset, derived from this process, is referred to as the instance-aware upper bound.

We denote $M_{S_i \rightarrow T_j}(x)$ as the output of model $M$ towards question $x$ after applying $\mathcal{X}$Transplant from $i^{\text{th}}$ layer of language $S$ to the $j^{\text{th}}$ layer of language $T$. Let $y_{true}$ represents the gold answer of question $x$ and $\mathbb{I}(\cdot)$ is a indicator function that equals 1 if the condition is true, 0 otherwise. The upper bound performance can be formulated as follows:

$$\text{UpperBound}_{\text{S}\rightarrow\text{T}}(M, D) = \sum_{x \in D} \max_{i,j \in \{1,...,N\}} \mathbb{I}(M_{S_i \rightarrow T_j}(x) = y_{true}) \tag{6}$$

Although enumerating all $N^2$ possibilities is inherently time-consuming, our goal is to benchmark the upper bound of the model's capabilities achievable through $\mathcal{X}$Transplant, and demonstrate the underlying mechanisms by enumerating all these results.

# 4 Experiments

## 4.1 Experimental Setup

**Models.** We selected 4 typical LLMs for our experiments and analysis. (1) *LLaMA-2-7B-Chat*, (2) *Mistral-7B-Instruct-v0.3*, (3) *Qwen2-7B-Instruct*, the three representative English-centric models are employed in our main experiments. (4) *Chinese-Alpaca-2-7B*, the alpaca-2 model further pretrained incrementally on large-scale Chinese data, are used for subsequent further analysis.

**Datasets.** We mainly conduct experiments on 4 benchmarks, which can be categorized into:

- **Multilingual Capability:** (1) *XNLI* (Conneau et al., 2018), a natural language inference corpus covering 15 languages, (2) *XQuAD* (Artetxe et al., 2020), a question answering dataset covering 12 languages, and (3) *XCOPA* (Ponti et al., 2020), a causal commonsense reasoning dataset covering 11 languages. These datasets consist of linguistically parallel questions designed to assess the model's ability across languages. For questions in non-English languages, we apply $\texttt{En} \rightarrow \texttt{non-En}$ $\mathcal{X}$Transplant to harness feed-forward activations from English.
- **Cultural Adaptability:** *GlobalOpinionQA* contains questions and answers from cross-national surveys designed to capture diverse opinions on global issues across different countries, all in English. This dataset aims to evaluate the model's cultural adaptability within an English context. For these questions in English, we apply $\texttt{non-En} \rightarrow \texttt{En}$ $\mathcal{X}$Transplant, hoping the model to leverage feed-forward activations from non-English languages to better capture cultural nuances.

Notably, due to the extensive scale of our experiments[1] , we did not use the full size of the datasets mentioned. For each dataset, we used `random.seed(666)` to randomly sample 50 instances in each language involved, creating our testbed. These smaller yet linguistically balanced datasets are referred to as *pilotsets*. Detailed information of each pilotset can be found in Appendix B.1.

---

[1]To obtain the instance-aware upper bound of $\mathcal{X}$Transplant, we perform inference on all $N^2$ possible source and target layer selection strategies for each instance in the dataset (for example, in *LLaMA-2-7B-Chat* with layer numbers $N = 32$, $N^2 = 1024$ times inference are conducted for each instance). Our main experiments involves 3 LLMs and 4 pilotset datasets, resulting in **over 800 hours** of computation on 8 * A800-SXM4-80GB.

Table 1: Main results on multilingual tasks. $\text{PIM}_{\text{En + lang}}$ denotes inputs with concatenated prompts in the involved language following the English version, while $\text{UpperBound}_{\text{En2lang}}$ represents $\mathcal{X}$Transplant from English to involved language.

| Models | Dataset: XNLI (PilotSet) | | | | | | | | | | | | | | | |
|---|---|---|---|---|---|---|---|---|---|---|---|---|---|---|---|---|
| | en | ar | bg | de | el | es | fr | hi | ru | sw | th | tr | ur | vi | zh | Avg |
| LLaMA-2-7B-Chat | 60.0 | 34.0 | 26.0 | 50.0 | 30.0 | 36.0 | 46.0 | 8.00 | 46.0 | 14.0 | 0.00 | 34.0 | 0.00 | 28.0 | 40.0 | 30.1 |
| $\text{PIM}_{\text{En + lang}}$ | 38.0 | 4.00 | 4.00 | 20.0 | 6.00 | 32.0 | 34.0 | 0.00 | 22.0 | 12.0 | 10.0 | 14.0 | 2.00 | 28.0 | 0.00 | 15.1 |
| Multilingual SFT | 30.0 | 38.0 | 28.0 | 36.0 | 62.0 | 32.0 | 36.0 | 32.0 | 44.0 | 16.0 | 34.0 | 30.0 | 8.00 | 38.0 | 34.0 | 33.2 |
| $\text{UpperBound}_{\text{En2lang}}$ | **94.0** | **90.0** | **96.0** | **100** | **96.0** | **84.0** | **100** | **60.0** | **98.0** | **82.0** | **66.0** | **74.0** | **34.0** | **84.0** | **100** | **83.9** |
| Mistral-7B-Instruct-v0.3 | 46.0 | 6.00 | 56.0 | 50.0 | 40.0 | 60.0 | 48.0 | 30.0 | 52.0 | 0.00 | 32.0 | 36.0 | 14.0 | 46.0 | 50.0 | 37.7 |
| $\text{PIM}_{\text{En + lang}}$ | 62.0 | 64.0 | 60.0 | 68.0 | 46.0 | 60.0 | 60.0 | 60.0 | 62.0 | 26.0 | 60.0 | 52.0 | 50.0 | 28.0 | 50.0 | 53.9 |
| Multilingual SFT | 42.0 | 44.0 | 36.0 | 34.0 | 56.0 | 44.0 | 40.0 | 40.0 | 50.0 | 4.00 | 24.0 | 28.0 | 38.0 | 48.0 | 40.0 | 37.9 |
| $\text{UpperBound}_{\text{En2lang}}$ | **80.0** | **72.0** | **64.0** | **76.0** | **98.0** | **78.0** | **82.0** | **84.0** | **78.0** | **36.0** | **88.0** | **82.0** | **66.0** | **78.0** | **92.0** | **76.9** |
| Qwen2-7B-Instruct | 82.0 | 52.0 | 54.0 | 56.0 | 52.0 | 68.0 | 70.0 | 50.0 | 64.0 | 26.0 | 48.0 | 50.0 | 32.0 | 60.0 | 64.0 | 55.2 |
| $\text{PIM}_{\text{En + lang}}$ | 84.0 | 70.0 | 72.0 | 54.0 | 48.0 | 72.0 | 62.0 | 62.0 | 72.0 | 56.0 | **76.0** | 10.0 | **78.0** | 58.0 | 62.0 | 62.4 |
| Multilingual SFT | 52.0 | 44.0 | 40.0 | 44.0 | 60.0 | 58.0 | 56.0 | 48.0 | 38.0 | 32.0 | 28.0 | 38.0 | 40.0 | 36.0 | 62.0 | 45.1 |
| $\text{UpperBound}_{\text{En2lang}}$ | **94.0** | **70.0** | **74.0** | **80.0** | **66.0** | **82.0** | **90.0** | **62.0** | **84.0** | **84.0** | 62.0 | **78.0** | 56.0 | **78.0** | **86.0** | **76.4** |

| Models | Dataset: XQuAD (PilotSet) | | | | | | | | | | | | |
|---|---|---|---|---|---|---|---|---|---|---|---|---|---|
| | en | ar | de | el | es | hi | ro | ru | th | tr | vi | zh | Avg |
| LLaMA-2-7B-Chat | 64.0 | 8.00 | 56.0 | 12.0 | 60.0 | 8.00 | 42.0 | 42.0 | 6.00 | 24.0 | 40.0 | 40.0 | 33.5 |
| $\text{PIM}_{\text{En + lang}}$ | 66.0 | 30.0 | 40.0 | 28.0 | 38.0 | **34.0** | 32.0 | 36.0 | 22.0 | 36.0 | 34.0 | 40.0 | 36.3 |
| Multilingual SFT | 24.0 | 52.0 | 28.0 | 68.0 | 72.0 | 12.0 | 54.0 | 46.0 | 20.0 | 42.0 | 42.0 | 56.0 | 43.0 |
| $\text{UpperBound}_{\text{En2lang}}$ | **92.0** | **34.0** | **80.0** | **38.0** | **84.0** | 32.0 | **74.0** | **82.0** | **30.0** | **64.0** | **66.0** | **70.0** | **62.2** |
| Mistral-7B-Instruct-v0.3 | 64.0 | 38.0 | 42.0 | 20.0 | 54.0 | 32.0 | 48.0 | 44.0 | 20.0 | 38.0 | 40.0 | 38.0 | 39.8 |
| $\text{PIM}_{\text{En + lang}}$ | 68.0 | 34.0 | 52.0 | 30.0 | 52.0 | 40.0 | 52.0 | 46.0 | 30.0 | 46.0 | 54.0 | 50.0 | 46.2 |
| Multilingual SFT | 38.0 | 52.0 | 28.0 | 70.0 | 56.0 | 28.0 | 46.0 | 48.0 | 30.0 | 40.0 | 40.0 | 56.0 | 44.3 |
| $\text{UpperBound}_{\text{En2lang}}$ | **90.0** | **54.0** | **76.0** | **50.0** | **78.0** | **50.0** | **80.0** | **72.0** | **50.0** | **68.0** | **66.0** | **76.0** | **67.5** |
| Qwen2-7B-Instruct | 76.0 | 52.0 | 40.0 | 22.0 | 48.0 | 18.0 | 36.0 | 48.0 | 38.0 | 46.0 | 64.0 | 80.0 | 47.3 |
| $\text{PIM}_{\text{En + lang}}$ | 68.0 | 42.0 | 50.0 | 20.0 | 50.0 | 32.0 | 44.0 | 48.0 | 38.0 | 46.0 | 60.0 | 66.0 | 47.0 |
| Multilingual SFT | 60.0 | 58.0 | 30.0 | 82.0 | 56.0 | 42.0 | 48.0 | 66.0 | 56.0 | 52.0 | 70.0 | 80.0 | 58.3 |
| $\text{UpperBound}_{\text{En2lang}}$ | **94.0** | **76.0** | **78.0** | **52.0** | **78.0** | **58.0** | **76.0** | **82.0** | **64.0** | **78.0** | **90.0** | **94.0** | **76.7** |

| Models | Dataset: XCOPA (PilotSet) | | | | | | | | | | | |
|---|---|---|---|---|---|---|---|---|---|---|---|---|
| | en | et | ht | id | it | sw | ta | th | tr | vi | zh | Avg |
| LLaMA-2-7B-Chat | 60.0 | 44.0 | 10.0 | 50.0 | 30.0 | 0.00 | 0.00 | 54.0 | 46.0 | 58.0 | 56.0 | 37.1 |
| $\text{PIM}_{\text{En + lang}}$ | 58.0 | 0.00 | 0.00 | 0.00 | 6.00 | 0.00 | 0.00 | 30.0 | 84.0 | 38.0 | 0.00 | 19.6 |
| Multilingual SFT | 66.0 | 56.0 | 40.0 | 54.0 | 50.0 | 50.0 | 30.0 | 18.0 | 54.0 | 16.0 | 46.0 | 43.6 |
| $\text{UpperBound}_{\text{En2lang}}$ | **94.0** | **58.0** | **60.0** | **100** | **100** | **54.0** | **60.0** | **56.0** | **100** | **78.0** | **100** | **78.2** |
| Mistral-7B-Instruct-v0.3 | 40.0 | 22.0 | 56.0 | 66.0 | 72.0 | 16.0 | 0.00 | 56.0 | 54.0 | 70.0 | 70.0 | 47.5 |
| $\text{PIM}_{\text{En + lang}}$ | 70.0 | 66.0 | 78.0 | 78.0 | 88.0 | 0.00 | **66.0** | 72.0 | 78.0 | 86.0 | **84.0** | 69.6 |
| Multilingual SFT | 82.0 | 56.0 | 36.0 | 70.0 | 80.0 | 14.0 | 48.0 | 60.0 | 48.0 | 40.0 | 70.0 | 54.9 |
| $\text{UpperBound}_{\text{En2lang}}$ | **94.0** | **76.0** | **92.0** | **88.0** | **92.0** | **54.0** | 28.0 | **72.0** | **80.0** | **86.0** | 74.0 | **76.0** |
| Qwen2-7B-Instruct | 0.00[2] | 44.0 | 52.0 | 86.0 | 88.0 | 62.0 | 36.0 | 50.0 | 28.0 | 90.0 | 84.0 | 56.4 |
| $\text{PIM}_{\text{En + lang}}$ | 6.00 | 6.00 | 72.0 | 0.00 | 38.0 | 36.0 | 70.0 | 24.0 | 48.0 | 0.00 | 26.0 | 29.6 |
| Multilingual SFT | 0.00 | 8.0 | 42.0 | 82.0 | 92.0 | 38.0 | 40.0 | 80.0 | 42.0 | 80.0 | 82.0 | 53.3 |
| $\text{UpperBound}_{\text{En2lang}}$ | **90.0** | **98.0** | **94.0** | **94.0** | **100** | **88.0** | **100** | **90.0** | **94.0** | **96.0** | **98.0** | **94.7** |

**Evaluations & Hyperparameters.** We evaluate all benchmarks using accuracy metric (details in Appendix B.2). And as mentioned in Section 3.1, $\mathcal{X}$Transplant transplants a single layer only when predicting the first new token. We use greedy decoding with a max of 20 new tokens for each model.

**Comparative Setup.** We compare the UpperBound performance achieved by $\mathcal{X}$Transplant with (1) the original performance of LLMs, (2) PIM (Mu et al., 2024), which concatenates prompts in two languages to activate more neurons and enhance multilingual potential, (3) CoT (Wei et al., 2022b) (results are provided in Appendix B.4), which prompts the models with step-by-step reasoning to unlock its potential, and (4) Multilingual SFT, which boosts multilingual capabilities by additional multilingual supervised fine-tuning. The implementation details are in Appendix B.3.

## 4.2 MAIN RESULTS

The main results of the multilingual datasets are presented in Table 1 and the results for the cultural dataset are illustrated in Figure 2. Notably, while we presented both the baseline results and the upper bound results of $\mathcal{X}$Transplant in the same table or figure, our goal is not to demonstrate the superiority of $\mathcal{X}$Transplant. Instead, we aim to use these comparisons to **illustrate the extent to which multilingual capabilities can be unlocked through the $\mathcal{X}$Transplant mechanism without modifying LLM itself**. The following are our observations:

---

[2]The explanation of accuracy in English subset of XCOPA for Qwen2-7B-Instruct is in Appendix B.5.

**(1) Underutilization of current LLMs' multilingual potential.** The results show that the upper-bound performance of $\mathcal{X}$Transplant is mostly much higher than the LLMs' original performance and even applying `PIM` method by simply concatenating multilingual prompts can result in over a 15% improvement on datasets like *XNLI* and *XCOPA*, as observed with *Mistral-7B-Instruct-v0.3*. These results all clearly indicate that the performance of the three representative English-centric LLMs is not fully realized in multilingual or culture-aware tasks, yet they hold significant potential for breakthroughs with some interventions.

**(2) Surprisingly high upper bound performance achievable through $\mathcal{X}$Transplant mechanism.** Through an exhaustive search within the $N^2$ answer space of $\mathcal{X}$Transplant, we established the upper bound performance of $\mathcal{X}$Transplant using Equation 6, as shown in Table 1 and Figure 2. Compared to the original performance of involved LLMs, $\mathcal{X}$Transplant exhibits surprisingly high upper bounds with a average relative increase of +80% in multilingual tasks and +39% in culture-aware task, which indicates that the LLMs' multilingual capability and cultural adaptability hold the potential to be significantly enhanced with the intervention of feed-forward activations from other languages. And the comparison with `Multilingual SFT` shows that the cross-lingual latent representation interaction enabled by $\mathcal{X}$Transplant not only offers substantial benefits but also has a strong chance of surpassing the improvements achieved through additional supervised fine-tuning, demonstrating an innovative and highly promising direction for extending the boundaries of LLM performance.

**(3) Improvements under English2English setting.** In Table 1, we observe that $\mathcal{X}$Transplant also yields performance gain under the English2English setting, which seems inconsistent with the idea that the benefits of $\mathcal{X}$Transplant stem from cross-lingual interactions. However, this result is logical. In this setting, $\mathcal{X}$Transplant simplifies to replacing the feed-forward activations between different decoder layers within the same input. Since different decoder layers of LLMs capture distinct features of the input and activate different neurons (i.e., knowledge), the transplanting operation between these layers can **strengthen feature propagation** and **encourage feature reuse**, leading to performance improvements. This phenomenon is analogous to the dense connections in DenseNet (Huang et al., 2017), which has been shown to enhance feature flow and overall performance.

**(4) English boosts multilingual capability, while non-English improves cultural adaptability.** $\mathcal{X}$Transplant supports transplantation in two directions: `En → non-En` for multilingual tasks and `non-En → En` for multicultural task. The results underscore the effectiveness of $\mathcal{X}$Transplant in both aspects, demonstrating that the feed-forward activations from English tend to strengthen the model's multilingual generalization, while feed-forward activations from non-English allow for deeper understanding of culturally specific content. This dual attempt reveals the complementary strengths of English and non-English activations in optimizing multilingual and cultural tasks.

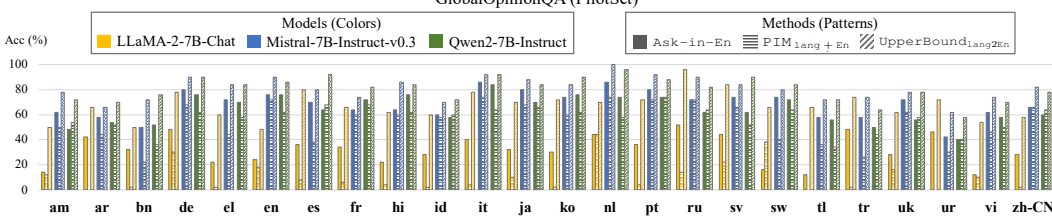

Figure 2: Main results on culture-aware task. Colors distinguish different LLMs, while patterns indicate applied methods. $\text{PIM}_{\text{lang + En}}$ concatenates the non-English prompt followed by English, while $\text{UpperBound}_{\text{lang2En}}$ represents $\mathcal{X}$Transplant from non-English language to English. The results of `CoT` and `Multilingual SFT` can be found in Figure 8 and Figure 9.

## 5 FURTHER ANALYSIS

In this section, we delve deeper into $\mathcal{X}$Transplant through a series of targeted analysis, which aim to solidify the foundational guarantees of $\mathcal{X}$Transplant and explore its broader practical applications. Additional analysis about $\mathcal{X}$Transplant outcomes can be found in the Appendix C.1.

## 5.1 ANALYSIS OF INPUT AND OUTPUT LANGUAGE CONSISTENCY

Ensuring consistency between input and output languages is a crucial capability for language models, which requires models to maintain the same language for both input and output unless explicitly instructed otherwise. $\mathcal{X}$Transplant improves LLMs' multilingual capability and cultural adaptability by transplanting feed-forward activations from inputs in other languages. To investigate whether these feed-forward activations might cause shifts in the output language, we performed a language consistency analysis on all answers within the $N^2$ answer space of $\mathcal{X}$Transplant.

The language consistency results[3] shown in Table 2 demonstrate that, the PIM method, leveraging multilingual contexts, is likely to result in inconsistency between the input and output languages. But under $\mathcal{X}$Transplant setting, the input-output language consistency across all answers in the $N^2$ answer space of $\mathcal{X}$Transplant approaches, and in some cases even exceeds, the consistency observed in the original setting. This indicates that feed forward activations from other languages rarely affect the language consistency during inference, making language shifts unlikely. This also provides a foundational guarantee for the upper bound results in Section 4.

Table 2: The input-output language consistency results of three LLMs with PIM and $\mathcal{X}$Transplant, compared with their original language consistency. non-En and En represent the input-output language required by corresponding tasks.

| Language Consistency (%) | XNLI (non-En) | XQuAD (non-En) | XCOPA (non-En) | GlobalOpinionQA (En) |
|---|---|---|---|---|
| LLaMA-2-7B-Chat | 95.20 | 83.00 | 86.93 | 99.83 |
| —— PIM | 59.75 | 77.05 | 84.51 | 89.35 |
| —— $\mathcal{X}$Transplant | 95.23 | 88.21 | 93.69 | 99.74 |
| Mistral-7B-Instruct-v0.3 | 88.13 | 91.83 | 84.91 | 100.0 |
| —— PIM | 63.07 | 86.67 | 85.45 | 90.75 |
| —— $\mathcal{X}$Transplant | 94.36 | 96.50 | 85.95 | 99.97 |
| Qwen2-7B-Instruct | 95.20 | 99.50 | 88.36 | 100.0 |
| —— PIM | 91.23 | 96.67 | 77.55 | 97.10 |
| —— $\mathcal{X}$Transplant | 97.43 | 99.22 | 87.09 | 99.92 |

## 5.2 ANALYSIS OF GENERALIZABILITY FROM ENGLISH- TO CHINESE-CENTRIC LLM

Our main experiments mainly focus on several representative English-centric LLMs, revealing that feed-forward activations from English can help enhance the model's multilingual capability. In this section, we further explore the generalizability of this conclusion by comparing the upper bound performance of $\mathcal{X}$Transplant on *LLaMA-2-7B-Chat* and *Chinese-Alpaca-2-7B*, a model based on LLaMA-2-7B that underwent further instruction-following fine-tuning and secondary pre-training with Chinese data.

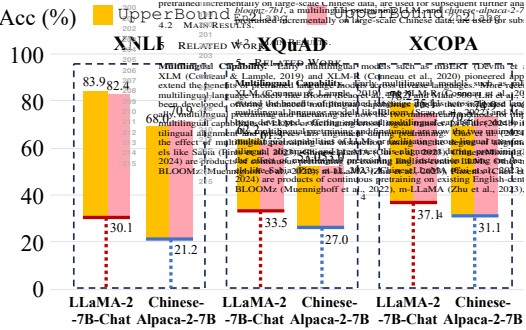

**Not only activations from English can help.** As shown in Figure 3, we find that for both English- and Chinese-centric LLMs, the feed-forward activations from either English or Chinese results in upper bound result that far exceeds the LLMs' original performance. This

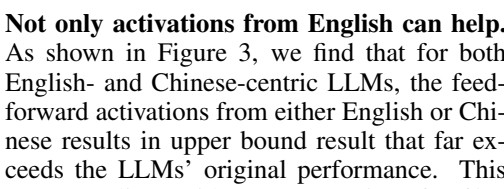

Figure 3: The upper bound results of English- and Chinese-centric LLM achieved by $\mathcal{X}$Transplant from English (UpperBound$_{En2lang}$) and Chinese (UpperBound$_{zh2lang}$). The horizontal line represents the model's original performance.

outcome aligns with our expectations for $\mathcal{X}$Transplant, as it leverages the knowledge activated in one language to assist another, without being confined to English as the sole source language.

**Native language preference in Native-centric LLMs.** Figure 3 further reveals that, for *LLaMA-2-7B-Chat*, the English-centric LLM, activations from English result in a higher upper bound in $\mathcal{X}$Transplant than those from Chinese (En: 74.8%, Zh: 72.9% in average). Meanwhile, in *Chinese-Alpaca-2-7B*, the Chinese-centric LLM, activations from Chinese can offer greater improvements (En: 63.7%, Zh: 66.3% in average). This indicates a preference in native-centric models, where feed-forward activations from the model's native language tend to yield more substantial gains, likely because the model's internal knowledge is more closely aligned with its native language.

---

[3]The languages are identified by *lid.176.bin* model from *fasttext*, which can recognize 176 languages.

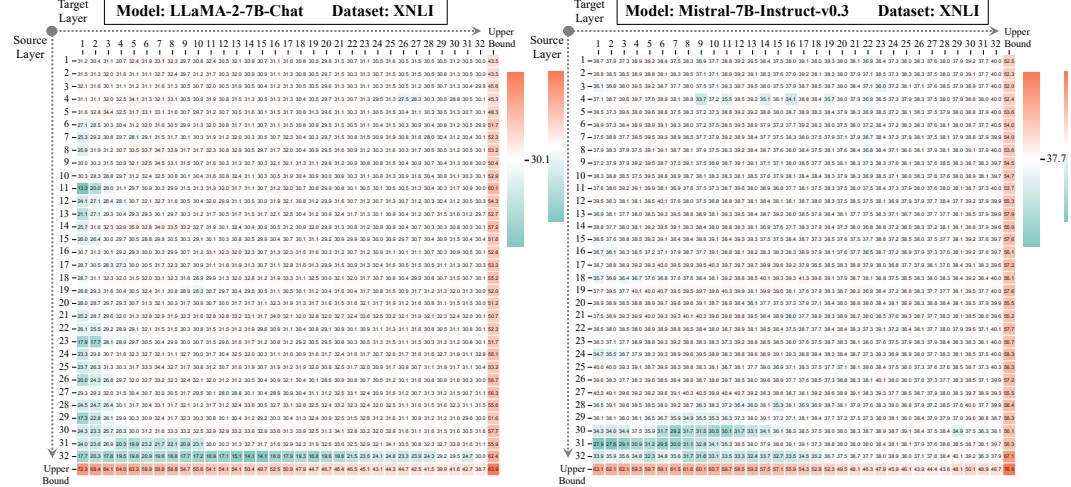

Figure 4: Accuracy results of $\mathcal{X}$Transplant acrosss all $N^2$ source and target layer selection strategies, along with the layer-specific upper bound performance obtained from a $N$ size answer space where either the source or target layer is fixed. The median in the legend represents the model's original performance; thus, red indicates better performance, while blue indicates worse performance.

## 5.3 ANALYSIS OF LAYER-SPECIFIC EFFECTIVENESS IN SOURCE AND TARGET SELECTION

In $\mathcal{X}$Transplant, the selection of source and target layers for the transplantation operation is undoubtedly a critical issue. The upper bound results in our main experiments were obtained by exploring all possible combinations of $N^2$ source and target layer. However, for practical application of $\mathcal{X}$Transplant, an instance-level dynamic strategy for selecting source and target layers is essential. Therefore, we also analyze the impact of different source and target layer selections on $\mathcal{X}$Transplant. Part of the results are illustrated in Figure 4, while additional results across more models and datasets can be found in the Appendix C.2. The layer-specific upper bounds, where either the source or target layer is fixed, are also presented in the figures.

**Limited improvement with fixed layer selections: The necessity of an instance-aware strategy.** As shown in Figure 4, while some minor improvements can be observed under certain settings, fixed strategies for selecting source and target layers generally do not yield satisfactory results. This underscores the necessity for an instance-aware strategy, where appropriate source and target layers are selected for each instance, to approach or even achieve the overall upper bound performance.

**Activations from the last layer provide the greatest benefit, and applying $\mathcal{X}$Transplant earlier in the target language yields higher upper bound.** From the perspective of source layer selection, the results in Figure 4 demonstrate that, though selecting the last layer as the source for $\mathcal{X}$Transplant often results in lower accuracy compared to the model's original performance, it leads to the highest upper bound. This suggests that the last layer likely contains significant multilingual knowledge beneficial to $\mathcal{X}$Transplant, but on an instance-level, the effectiveness of these activations largely depends on the selection of the target layer. From the perspective of target layer selection, we found that applying $\mathcal{X}$Transplant earlier in the inference stage of the target language results in a higher upper bound performance. This might be because it allows sufficient forward propagation space for the model to integrate knowledge from other languages, rather than having knowledge from other languages dominate at the final stages of inference.

**Narrowing the $N^2$ search space to $N$.** For practical application of $\mathcal{X}$Transplant on each instance, selecting the appropriate source layer and target layer from the $N^2$ choice space seems challenging. However, the layer-specific results provide two alternative strategies: either fix the source layer as the last layer and select the target layer from $N$ options, or fix the target layer as the first layer and choose the source layer from $N$ options. These two approaches significantly reduce the complexity of source and target layer selection from $N^2$ to $N$ while still approaching the overall upper bound performance (original: 45.7%, source-last: 66.3%, target-first:68.2%, overall: 76.8% in average).

# 6 DISCUSSION: UNPACKING WHAT IS BEHIND $\mathcal{X}$TRANSPLANT

## 6.1 $\mathcal{X}$TRANSPLANT IS A RELIABLE AND STABLE ACTIVATION MODIFICATION MECHANISM

From the perspective of language modeling, the essence of "$\mathcal{X}$Transplant affect model's output" lies in the fact that certain intermediate states within the model's inference process are altered, which in turn affects the probability distribution of the next token prediction. This mechanism is fundamentally similar to some approaches in fields such as *model editing* (Yao et al., 2023; Zhang et al., 2024) and *controllable text generation* (Liang et al., 2024; Konen et al., 2024).

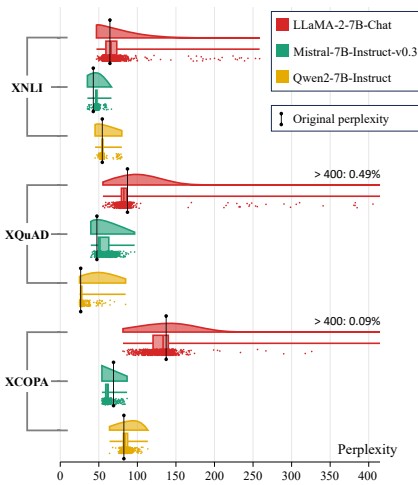

Generally, directly modifying activations within a model's inference process is a delicate operation that, if not handled carefully, can easily cause the model's output to break down. However, while inputs in different languages may present linguistic differences, they still share commonalities as they stem from the same question being input in the same model. $\mathcal{X}$Transplant skillfully exploits both these differences and commonalities, allowing the model to benefit from the broader multilingual knowledge (differences) while ensuring that the feed-forward activations from other languages remain compatible and do not disrupt the model's output (commonalities). The results in Figure 5, showing the perplexity distribution under all $N^2$ transplantation strategies alongside the model's original average perplexity, demonstrate $\mathcal{X}$Transplant's reliability and stability (see details in Appendix C.3). Moreover, $\mathcal{X}$Transplant limits the modification of intermediate activations to $N^2$ possible choices (or even narrows it down to $N$, as discussed in Section 5.3), which, compared to making arbitrary changes to hidden states, ensures that the impact of $\mathcal{X}$Transplant on the model's output remains more stable and relatively controllable.

Figure 5: The perplexity distribution under all $N^2$ $\mathcal{X}$Transplant strategies across different LLMs and datasets, compared with the models' original perplexity results.

## 6.2 A CASE STUDY: FROM THE PERSPECTIVE OF INTERMEDIATE DECODING

To further understand how $\mathcal{X}$Transplant alters the model's output step by step, we present a real case study in Figure 6 in a more interpretable way of intermediate decoding.

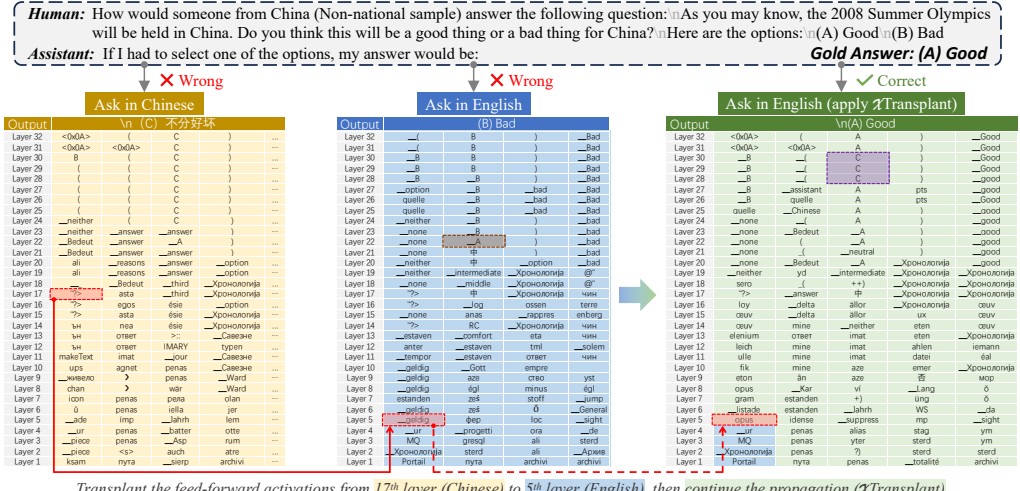

Figure 6: A intermediate decoding case study of transplanting the feed forward activations from Chinese to English, compared with its original responses when prompting in Chinese and English.

The example question in Figure 6 is a real case from the *GlobalOpinionQA* dataset, with all responses generated by *LLaMA-2-7B-Chat*. We present the model's responses for the *Ask-in-Chinese* prompt, *Ask-in-English* prompt, and a response selected from the $N^2$ answer space of $\mathcal{X}$Transplant from Chinese to English. As shown, when prompted in Chinese, *LLaMA-2-7B-Chat*, due to its limited proficiency in Chinese, produced a hallucinated response (C) that was not among the given answer options. When prompted in English, *LLaMA-2-7B-Chat* also provided an incorrect answer (B). However, by checking the intermediate decoding process of *Ask-in-English*, we found that *LLaMA-2-7B-Chat* had the potential to produce the correct answer, as highlighted in the **brown box**. By applying $\mathcal{X}$Transplant from the 17th layer (Chinese) to the 5th layer (English), the feed-forward activations from Chinese successfully guided the model to give the correct answer (A). Nevertheless, as highlighted in **purple box**, there is also a risk of over-guidance with $\mathcal{X}$Transplant, where knowledge from the source language may excessively influence the model's decision.

## 7 RELATED WORK

**Multilingual Capability.** Early multilingual models like mBERT (Devlin et al., 2019) and XLM (Conneau & Lample, 2019) laid the groundwork for extending pretrained models across diverse languages. Recently larger multilingual models, such as Bloom (Scao et al., 2022) and Mala-500 (Lin et al., 2024), enhance multilingual capabilities through increased scale. Generally, multilingual pretraining and finetuning are now the two mainstream methods for improving multilingual performance. Works like Li et al. (2024b) injects multilingual alignment and preserves this during pretraining. Gao et al. (2024) explored the effect of multilingual pretraining and instruction tuning on the degree of alignment. Models like Sabia (Pires et al., 2023), ChineseLLaMA (Cui et al., 2023), ChineseMixtral (HIT-SCIR, 2024) are products of continuous pretraining on existing English-centric LLMs. Other like BLOOMz (Muennighoff et al., 2022), m-LLaMA (Zhu et al., 2023), Phoenix (Chen et al., 2023) chosen to directly incorporate multilingual data in the supervised finetuning stage to achieve implicit multilingual alignment across languages.

**Cultural Adaptability.** Previous studies have shown that current LLMs exhibit poor cultural adaptability (Ramezani & Xu, 2023; Jha et al., 2023; Rao et al., 2024). Solutions towards these culture-aware challenges can be categorized mainly into two approaches: context learning and training-based. Kovač et al. (2023) studied models' controllability in inducing cultural perspectives, while Wang et al. (2024) improved cultural performance by explicitly prompting LLMs with the recognition of culture in queries. Rao et al. (2023) developed a framework integrating moral dilemmas with principles from various normative ethics formalisms across different levels of abstraction. Rao et al. (2023) developed a framework integrating ethics from diverse cultures. Another line of research involves fine-tuning models on large-scale culturally relevant datasets (Abbasi et al., 2023; Lin & Chen, 2023; Nguyen et al., 2024; Shi et al., 2024), or investing in more balanced multilingual corpus for pretraining (Scao et al., 2022; Lin et al., 2024; Gao et al., 2024; Li et al., 2024b).

Unlike previous training-based approaches, $\mathcal{X}$Transplant directly modifies the model's internal activations during inference, allowing the model to benefit from both English and non-English inputs. This simple yet promising mechanism marks a new step forward in cross-lingual capability transfer.

## 8 CONCLUSION

In this work, we introduce $\mathcal{X}$Transplant, a probing method that contributes to further unlocking the multilingual potential of LLMs, as well as their cultural adaptability, by cross-lingual feed-forward activations transplantation. Our extensive experiments across four representative LLMs and four datasets, along with established upper-bound performance, highlight the underutilization of current LLMs' multilingual capabilities. Besides that, we find that the feed-forward activations from English can significantly enhance the model's multilingual performance, while those from non-English languages enable a deeper and more nuanced understanding of culturally specific content. Our additional analysis and discussion further underscore the practical applicability of $\mathcal{X}$Transplant. Overall, our study not only introduces a novel probing method to improve LLMs' multilingual performance but also offers a deeper understanding of the mechanisms underlying multilingual knowledge transfer. We hope that $\mathcal{X}$Transplant will serve as a catalyst for future research, driving continued progress in developing more linguistically effective and culturally aware language models.

REPRODUCIBILITY

For better reproducibility, our implementation code of $\mathcal{X}$Transplant as well as the evaluation scripts are provided as the supplemental materials. And our pilotset versions of *XNLI*, *XQuAD*, *XCOPA* and *GlobalOpinionQA* datasets are also available with our code as the testbed for other researchers.

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

## A    POTENTIAL QUESTIONS AND EXPLANATIONS

1. **What's the reason for applying $\mathcal{X}$Transplant only when generating the first new token?**

   In autoregressive generation, applying $\mathcal{X}$Transplant during the generation of the first new token essentially introduces the benefit of feed-forward activations from another language across the entire sequence generation process. This is because all subsequent tokens are influenced by the activations cached from earlier steps. If $\mathcal{X}$Transplant were applied during the generation of every token, it would be a redundant operation and could even cause the model's output to break down.

2. **Why apply `En → non-En` $\mathcal{X}$Transplant in multilingual tasks and `non-En → En` in culture-aware tasks?**

   For the multilingual datasets (*XNLI*, *XQuAD*, and *XCOPA*), all of the questions are linguistically parallel across languages. These datasets assess the model's multilingual capabilities by asking questions in various languages such as Chinese, Spanish, German, French, etc. When posing questions in these non-English languages, we aim for the model to benefit from feed-forward activations derived from English. Therefore, for multilingual tasks, we perform `En → non-En` $\mathcal{X}$Transplant, where questions are asked in non-English languages, and activations from English are transplanted to the non-English languages.

   Regarding the culture-aware dataset, *GlobalOpinionQA*, all the questions and answers are in English. The purpose of this dataset is to explore how well models respond to questions from different cultural backgrounds within an English context. When asking questions in English, we want the model to leverage feed-forward activations from non-English languages to better capture cultural nuances. Hence, for culture-aware tasks, we perform `non-En → En` $\mathcal{X}$Transplant, where the questions are in English, but activations from non-English languages are transplanted into the English context. For example, when asking a question related to Chinese culture, we ask the question in English but feed-forward activations from Chinese are transplanted to help.

3. **Concern about "the computational cost".**

   It is important to emphasize that the extensive experiments serve to demonstrate the potential of $\mathcal{X}$Transplant as a simple yet promising mechanism for enhancing multilingual capabilities and cross-lingual transfer, without any modifications to the LLM itself.

   As the first work to explore this mechanism, we felt that the effort spent on a comprehensive analysis was justified, and we believe it has successfully showcased the substantial potential of $\mathcal{X}$Transplant.

   Additionally, while our work involves significant computational expense, **we do not intend for future research to replicate this approach in a resource-intensive manner**. Our goal is to inspire future work that can utilize the concept/mechanism of cross-lingual feed-forward activations transplantation to model design or training phases, such as exploring cross-lingual feed-forward interactions and connections between different layers or language-specific regions, to enable the model better leverage its intrinsic multilingual knowledge.

   Furthermore, the idea of transplantation is not limited to multilingual or cross-lingual scenarios but could also be applied to cross-task or cross-domain settings.

   Lastly, we hope that the large-scale experiments we conducted, along with our subsequent analysis and discussions, can inspire future research and provide valuable insights for the broader research community.

## B  Experimental Details

### B.1  Datasets

Due to the extensive scale of our experiments, we did not use the full version of each dataset. Instead, we conducted our experiments on pilotsets from each dataset. Specifically, each pilotset was obtained by randomly sampling 50 examples from the samples in each language covered by the full dataset, with the random seed set to `random.seed(666)`. For better reproducibility, these pilotsets will be publicly available along with our code. The detailed information of these pilotsets is as follows:

**Involved Languages**

XNLI (15): ar, bg, de, el, en, es, fr, hi, ru, sw, th, tr, ur, vi, zh
XQuAD (12): ar, de, el, en, es, hi, ro, ru, th, tr, vi, zh
XCOPA (11): en, et, ht, id, it, sw, ta, th, tr, vi, zh
GlobalOpinionQA (24): am, ar, bn, de, el, en, es, fr, hi, id, it, ja, ko, nl, pt, ru, sv, sw, tl, tr, uk, ur, vi, zh-CN

**Sample Size** (50 samples per language)

XNLI: $50 \times 15 = 750$
XQuAD: $50 \times 12 = 600$
XCOPA: $50 \times 11 = 550$
GlobalOpinionQA: $50 \times 24 = 1200$

### B.2  Evaluations

The prompts we used for each dataset are listed in Table 3. For each model involved, we apply greedy decoding strategy and set the max new tokens generated by the model to 20. We used Accuracy as our evaluation metric, and for different task types within each dataset, we applied the following rules:

- **For Multiple-choice Tasks (Classification):** *XNLI*, *XCOPA*, and *GlobalOpinionQA* all belong to the multiple-choice category. For these tasks, a model's response is considered correct only if it contains the correct option and excludes all other options.
- **For Question-Answering Tasks (Generation):** For the generative task *XQuAD*, the model's answer is deemed correct if the gold answer appears in the model's response.

To ensure better reproducibility, these evaluation scripts will also be made publicly available.

### B.3  Comparative Setup

In our main experiments, we compared the upper bound performance achieved by $\mathcal{X}$Transplant with the models' original performance, PIM (**P**arallel **I**nput in **M**ultiple Languages) (Mu et al., 2024) and CoT (**C**hain **o**f **T**hought) (Wei et al., 2022b). Below, we provide a detailed introduction to the implementations.

- **Multilingual Capability:** For multilingual datasets *XNLI*, *XQuAD*, and *XCOPA*: (1) The models' original performance refers to the performance when the same question is asked in different languages. (2) PIM$_{En + lang}$ concatenates prompt non-English language following the English version prompt, with the intention of prompting the model to output responses in corresponding non-English language. (3) CoT prompts the models with the suffix of "Let's think step by step" to utilize their further potential. (4) UpperBound$_{En2lang}$ represents the upper bound performance achieved by $\mathcal{X}$Transplant when transplanting feed-forward activations from English into other languages.
- **Cultural Adaptability:** For the *GlobalOpinionQA* dataset, which is designed to assess cultural adaptability in an English-speaking context, both the input and output languages are English. (1) The models' original performance refers to how well the model answers questions related to different cultural backgrounds. (2) PIM$_{lang + En}$ concatenates the English version of the prompt after prompts in other non-English language, aiming to have the model continue generating responses in English. (3) CoT prompts the models with the suffix of "Let's think step by step"

to utilize their further potential. (4) UpperBound $_{lang2En}$ represents the upper bound performance achieved by $\mathcal{X}$Transplant when transplanting feed-forward activations from non-English languages into English.

- **Detailed implementation of** Multilingual SFT**:** We randomly selected a total of 20,236 multilingual instruction pairs from *aya dataset* (Singh et al., 2024), ensuring language balance, and performed multilingual supervised fine-tuning on our involved three LLMs. The training was conducted on 8 A800-SXM4-80GB with the following settings: batch size=16, epochs=3, learning rate=1.0e-5, warmup ratio=0.1, and bf16=true.

### B.4 CHAIN-OF-THOUGHT RESULTS

As a prominent approach to further utilize LLMs, Chain-of-Thought has been examined in our work for its performance in multilingual and culture-aware tasks, with results illustrated in Table 4 and Figure 8.

The results indicate that CoT does not appear to be an effective method for further unlocking the model's potential in multilingual and culture-aware scenarios.

### B.5 EXPLANATION OF ACCURACY IN ENGLISH SUBSET OF XCOPA FOR QWEN2-7B-INSTRUCT

In Table 1, we notice that the accuracy in the English subset of XCOPA for *Qwen2-7B-Instruct* is "0.00". After specifically revisiting *Qwen2-7B-Instruct*'s responses to the English subset of XCOPA. We found that the "0.00 accuracy" issue stemms from the model's failure to effectively follow the instructions in our prompt. The exact prompt we used was:

> You are assigned to complete a two-category classification task.
>
> Premise: The girl squeezed her nose.
> Options: (1) The baby drools on the bib.
> (2) The baby soiled his diaper.
>
> Please determine which of the two options is more likely to be the cause of the given premise.
>
> Your Answer:

However, *Qwen2-7B-Instruct*'s responses are as follows:

> Option 1 (The baby drools on the bib) is less likely to be the cause of ...
> Option 1, "The audience clapped their hands to the music," is more likely to be ...
> Option 1 is more likely to be the result of the given premise. If the man expected the ...
> Option 2, "Her opponent felt sorry for her," is more likely to be the result of ...
> Option 2, The products are made by child labor. \n\n Explanation: The premise states that radicals ...
> Option 2, "It's snack time," is more likely to be the cause of the given ...
> ...

Our evaluation script for *XCOPA* dataset considers a model's response correct only if it contains the correct option (e.g., (1) or (2)) and excludes all other options. But as you can see above, Qwen-2's responses do not match this format, leading to the "0.0 accuracy".

To ensure fairness in evaluation, we can not arbitrarily modify our evaluation script based solely on Qwen's responses on the English subset of the *XCOPA* dataset. Therefore, we have retained this result in our main experimental table.

## C ANALYSIS

### C.1 PROPORTION ANALYSIS OF $\mathcal{X}$TRANSPLANT OUTCOMES

To further understand $\mathcal{X}$Transplant, for each question in the datasets, we analyzed the model's performance in three scenarios: whether it answered correctly in the source language, in the target language, and whether a correct answer exists in the $N^2$ answer space after applying $\mathcal{X}$Transplant

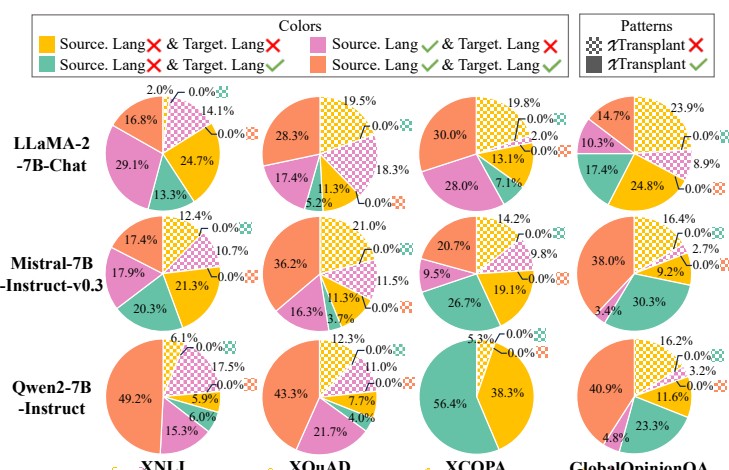

Figure 7: Proportion of all $\mathcal{X}$Transplant outcomes across 8 correctness categories. ✓ and × represent whether the model answered correctly or not under given settings.

from the source language to the target language. The combination of correctness in these three settings results in 8 distinct categories. In Figure 7, we present the sample proportions for these eight categories across three models and four datasets, leading to the following conclusions:

**$\mathcal{X}$Transplant does not introduce additional mistakes.** The results in Figure 7 across three models and four datasets consistently indicate that for questions that the model could correctly answer in the target language (i.e., the language which received feed forward activations from others), a correct answer is always present in the answer space after applying $\mathcal{X}$Transplant, as the corresponding proportions all being 0%. This reveals that when $\mathcal{X}$Transplant is appropriately utilized, it essentially serves as an enhancement strategy that does not impair the model's original performance.

**$\mathcal{X}$Transplant benefits more when the question can be accurately answered in source language.** The results in Figure 7 indicate that, in most cases, most of the questions that the model answers correctly using $\mathcal{X}$Transplant are those that could be correctly answered in the source language itself, regardless of correctness in target language. This demonstrates that feed-forward activations from a source language where the model can answer the question correctly help $\mathcal{X}$Transplant achieve better cross-lingual enhancement.

## C.2 LAYER-SPECIFIC ANALYSIS

In Section 5.3, due to the page limit, the figure 4 only present the layer-specific effectiveness results of *LLaMA-2-7B-Chat* and *Mistral-7B-Instruct-v0.3* on *XNLI*. Complete results across additional models and datasets can be found in Figure 11 and Figure 10.

## C.3 PERPLEXITY CALCULATION

The perplexity results in Section 6.1 include the average perplexity of the model under original conditions, as well as the average perplexity distribution across all $N^2$ settings of $\mathcal{X}$Transplant, encompassing 3 LLMs and 3 datasets. Notably, to mitigate the interference caused by overly short responses, we only included responses with a token length greater than 5 in our statistics.

Table 3: The prompts used for *XNLI*, *XQuAD*, *XCOPA* and *GlobalOpinionQA*.

---

Prompt for *XNLI* (English version)

---

Human: What do you think is the relationship between the premise and the hypothesis?

Premise: {premise}

Hypothesis: {hypothesis}

(1) Entail
(2) Neutral
(3) Contradict

Assistant: If I had to select one of the options, my answer would be: {response}

---

Prompt for *XQuAD* (English version)

---

Human: Please answer these questions only based on the given context.

Context: {context}

Question: {question}

Assistant: My answer would be: {response}

---

Prompt for *XCOPA* (English version)

---

You are assigned to complete a two-category classification task.

Premise: {premise}

Options: {options}

Please determine which of the two options is more likely to be the result of the given premise.

Your Answer: {response}

---

Prompt for *GlobalOpinionQA* (English version)

---

Human: How would someone from country answer the following question:
{question}
Here are the options:
{options}

Assistant: If I had to select one of the options, my answer would be: {response}

---

Table 4: Main results on multilingual tasks. CoT represents prompting models with a step-by-step reasoning process. $PIM_{En+lang}$ denotes inputs with concatenated prompts in the involved language following the English version, while $UpperBound_{En2lang}$ represents $\mathcal{X}$Transplant from English to involved language.

| Models | Dataset: XNLI (PilotSet) | | | | | | | | | | | | | | | |
|---|---|---|---|---|---|---|---|---|---|---|---|---|---|---|---|---|
| | en | ar | bg | de | el | es | fr | hi | ru | sw | th | tr | ur | vi | zh | Avg |
| LLaMA-2-7B-Chat | 60.0 | 34.0 | 26.0 | 50.0 | 30.0 | 36.0 | 46.0 | 8.00 | 46.0 | 14.0 | 0.00 | 34.0 | 0.00 | 28.0 | 40.0 | 30.1 |
| CoT | 36.0 | 34.0 | 26.0 | 32.0 | 10.0 | 24.0 | 18.0 | 28.0 | 10.0 | 18.0 | 14.0 | 12.0 | 0.00 | 20.0 | 8.00 | 19.3 |
| $PIM_{En+lang}$ | 38.0 | 4.00 | 4.00 | 20.0 | 6.00 | 32.0 | 34.0 | 0.00 | 22.0 | 12.0 | 10.0 | 14.0 | 2.00 | 28.0 | 0.00 | 15.1 |
| Multilingual SFT | 30.0 | 38.0 | 28.0 | 36.0 | 62.0 | 32.0 | 36.0 | 32.0 | 44.0 | 16.0 | 34.0 | 30.0 | 8.00 | 38.0 | 34.0 | 33.2 |
| $UpperBound_{En2lang}$ | **94.0** | **90.0** | **96.0** | **100** | **96.0** | **84.0** | **100** | **60.0** | **98.0** | **82.0** | **66.0** | **74.0** | **34.0** | **84.0** | **100** | **83.9** |
| Mistral-7B-Instruct-v0.3 | 46.0 | 6.00 | 56.0 | 50.0 | 40.0 | 60.0 | 48.0 | 30.0 | 52.0 | 0.00 | 32.0 | 36.0 | 14.0 | 46.0 | 50.0 | 37.7 |
| CoT | 62.0 | 22.0 | 42.0 | 50.0 | 0.00 | 60.0 | 32.0 | 8.00 | 34.0 | 24.0 | 36.0 | 26.0 | 4.00 | 22.0 | 18.0 | 29.3 |
| $PIM_{En+lang}$ | 62.0 | 64.0 | 60.0 | 68.0 | 46.0 | 60.0 | 60.0 | 60.0 | 62.0 | 26.0 | 60.0 | 52.0 | 50.0 | 28.0 | 50.0 | 53.9 |
| Multilingual SFT | 42.0 | 44.0 | 36.0 | 34.0 | 56.0 | 44.0 | 40.0 | 40.0 | 40.0 | 4.00 | 24.0 | 28.0 | 38.0 | 48.0 | 40.0 | 37.9 |
| $UpperBound_{En2lang}$ | **80.0** | **72.0** | **64.0** | **76.0** | **98.0** | **78.0** | **82.0** | **84.0** | **78.0** | **36.0** | **88.0** | **82.0** | **66.0** | **78.0** | **92.0** | **76.9** |
| Qwen2-7B-Instruct | 82.0 | 52.0 | 54.0 | 56.0 | 52.0 | 68.0 | 70.0 | 50.0 | 64.0 | 26.0 | 48.0 | 50.0 | 32.0 | 60.0 | 64.0 | 55.2 |
| CoT | 64.0 | 50.0 | 34.0 | 44.0 | 6.00 | 76.0 | 60.0 | 16.0 | 64.0 | 16.0 | 12.0 | 40.0 | 20.0 | 52.0 | 62.0 | 41.1 |
| $PIM_{En+lang}$ | 84.0 | 70.0 | 72.0 | 54.0 | 48.0 | 72.0 | 62.0 | 62.0 | 72.0 | 56.0 | **76.0** | 10.0 | **78.0** | 58.0 | 62.0 | 62.4 |
| Multilingual SFT | 52.0 | 44.0 | 40.0 | 44.0 | 60.0 | 56.0 | 56.0 | 44.0 | 38.0 | 32.0 | 28.0 | 38.0 | 40.0 | 36.0 | 62.0 | 45.1 |
| $UpperBound_{En2lang}$ | **94.0** | **70.0** | **74.0** | **80.0** | **66.0** | **82.0** | **90.0** | **62.0** | **84.0** | **84.0** | 62.0 | **78.0** | 56.0 | **78.0** | **86.0** | **76.4** |

| Models | Dataset: XQuAD (PilotSet) | | | | | | | | | | | | |
|---|---|---|---|---|---|---|---|---|---|---|---|---|---|
| | en | ar | de | el | es | hi | ro | ru | th | tr | vi | zh | Avg |
| LLaMA-2-7B-Chat | 64.0 | 8.00 | 56.0 | 12.0 | 60.0 | 8.00 | 42.0 | 42.0 | 6.00 | 24.0 | 40.0 | 40.0 | 33.5 |
| CoT | 68.0 | 8.00 | 48.0 | 14.0 | 56.0 | 8.00 | 36.0 | 34.0 | 0.00 | 18.0 | 38.0 | 32.0 | 30.0 |
| $PIM_{En+lang}$ | 66.0 | 30.0 | 40.0 | 28.0 | 38.0 | **34.0** | 32.0 | 36.0 | 22.0 | 36.0 | 34.0 | 40.0 | 36.3 |
| Multilingual SFT | 24.0 | 52.0 | 28.0 | 68.0 | 72.0 | 12.0 | 54.0 | 46.0 | 20.0 | 42.0 | 42.0 | 56.0 | 43.0 |
| $UpperBound_{En2lang}$ | **92.0** | **34.0** | **80.0** | **38.0** | **84.0** | 32.0 | **74.0** | **82.0** | **30.0** | **64.0** | **66.0** | **70.0** | **62.2** |
| Mistral-7B-Instruct-v0.3 | 64.0 | 38.0 | 42.0 | 20.0 | 54.0 | 32.0 | 48.0 | 44.0 | 20.0 | 38.0 | 40.0 | 38.0 | 39.8 |
| CoT | 72.0 | 12.0 | 64.0 | 4.00 | 64.0 | 16.0 | 54.0 | 42.0 | 8.00 | 24.0 | 16.0 | 48.0 | 35.3 |
| $PIM_{En+lang}$ | 68.0 | 34.0 | 52.0 | 30.0 | 52.0 | 40.0 | 52.0 | 46.0 | 30.0 | 46.0 | 54.0 | 50.0 | 46.2 |
| Multilingual SFT | 38.0 | 52.0 | 28.0 | 70.0 | 56.0 | 28.0 | 46.0 | 48.0 | 30.0 | 40.0 | 40.0 | 56.0 | 44.3 |
| $UpperBound_{En2lang}$ | **90.0** | **54.0** | **76.0** | **50.0** | **78.0** | **50.0** | **80.0** | **72.0** | **50.0** | **68.0** | **66.0** | **76.0** | **67.5** |
| Qwen2-7B-Instruct | 76.0 | 52.0 | 60.0 | 22.0 | 48.0 | 18.0 | 36.0 | 48.0 | 38.0 | 46.0 | 64.0 | 80.0 | 47.3 |
| CoT | 82.0 | 64.0 | 60.0 | 20.0 | 56.0 | 30.0 | 48.0 | 64.0 | 32.0 | 48.0 | 64.0 | 74.0 | 53.5 |
| $PIM_{En+lang}$ | 68.0 | 42.0 | 50.0 | 20.0 | 50.0 | 32.0 | 44.0 | 48.0 | 38.0 | 46.0 | 60.0 | 66.0 | 47.0 |
| Multilingual SFT | 60.0 | 58.0 | 30.0 | **82.0** | 56.0 | 42.0 | 48.0 | 66.0 | 56.0 | 52.0 | 70.0 | 80.0 | 58.3 |
| $UpperBound_{En2lang}$ | **94.0** | **76.0** | **78.0** | 52.0 | **78.0** | **58.0** | **76.0** | **82.0** | **64.0** | **78.0** | **90.0** | **94.0** | **76.7** |

| Models | Dataset: XCOPA (PilotSet) | | | | | | | | | | | |
|---|---|---|---|---|---|---|---|---|---|---|---|---|
| | en | et | ht | id | it | sw | ta | th | tr | vi | zh | Avg |
| LLaMA-2-7B-Chat | 60.0 | 44.0 | 10.0 | 50.0 | 30.0 | 0.00 | 0.00 | 54.0 | 46.0 | 58.0 | 56.0 | 37.1 |
| CoT | 60.0 | 14.0 | 2.00 | 36.0 | 24.0 | 14.0 | 6.00 | 14.0 | 44.0 | 40.0 | 54.0 | 28.0 |
| $PIM_{En+lang}$ | 58.0 | 0.00 | 0.00 | 0.00 | 6.00 | 0.00 | 0.00 | 30.0 | 84.0 | 38.0 | 0.00 | 19.6 |
| Multilingual SFT | 66.0 | 56.0 | 40.0 | 54.0 | 50.0 | 50.0 | 30.0 | 18.0 | 54.0 | 16.0 | 46.0 | 43.6 |
| $UpperBound_{En2lang}$ | **94.0** | **58.0** | **60.0** | **100** | **100** | **54.0** | **60.0** | **56.0** | **100** | **78.0** | **100** | **78.2** |
| Mistral-7B-Instruct-v0.3 | 40.0 | 22.0 | 56.0 | 66.0 | 72.0 | 16.0 | 0.00 | 56.0 | 54.0 | 70.0 | 70.0 | 47.5 |
| CoT | 36.0 | 16.0 | 6.00 | 58.0 | 52.0 | 0.00 | 2.00 | 2.00 | 2.0 | 28.0 | 38.0 | 21.8 |
| $PIM_{En+lang}$ | 70.0 | 66.0 | 78.0 | 78.0 | 88.0 | 0.00 | **66.0** | 72.0 | 78.0 | 86.0 | **84.0** | 69.6 |
| Multilingual SFT | 82.0 | 56.0 | 36.0 | 70.0 | 80.0 | 14.0 | 48.0 | 60.0 | 48.0 | 40.0 | 70.0 | 54.9 |
| $UpperBound_{En2lang}$ | **94.0** | **76.0** | **92.0** | **88.0** | **92.0** | **54.0** | 28.0 | **72.0** | **80.0** | **86.0** | 74.0 | **76.0** |
| Qwen2-7B-Instruct | 0.00 | 44.0 | 52.0 | 86.0 | 88.0 | 62.0 | 36.0 | 50.0 | 28.0 | 90.0 | 84.0 | 56.4 |
| CoT | 10.0 | 30.0 | 20.0 | 48.0 | 60.0 | 28.0 | 18.0 | 14.0 | 32.0 | 74.0 | 76.0 | 37.3 |
| $PIM_{En+lang}$ | 6.00 | 6.00 | 72.0 | 0.00 | 38.0 | 36.0 | 70.0 | 24.0 | 48.0 | 0.00 | 26.0 | 29.6 |
| Multilingual SFT | 0.00 | 8.0 | 42.0 | 82.0 | 92.0 | 38.0 | 40.0 | 80.0 | 42.0 | 80.0 | 82.0 | 53.3 |
| $UpperBound_{En2lang}$ | **90.0** | **98.0** | **94.0** | **94.0** | **100** | **88.0** | **100** | **90.0** | **94.0** | **96.0** | **98.0** | **94.7** |

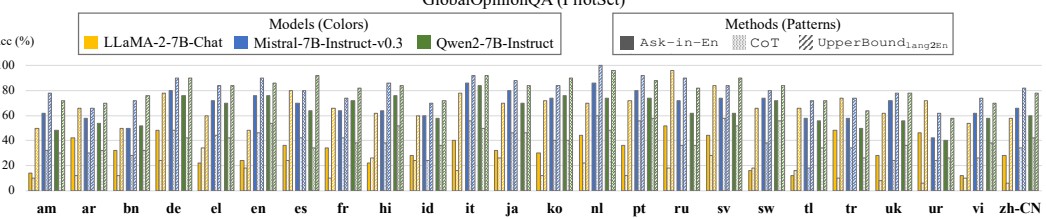

GlobalOpinionQA (PilotSet)

Figure 8: Main results on culture-aware task. Colors distinguish different LLMs, while patterns indicate applied methods. CoT represents prompting models with a step-by-step reasoning process, while $UpperBound_{lang2En}$ represents $\mathcal{X}$Transplant from non-English language to English.

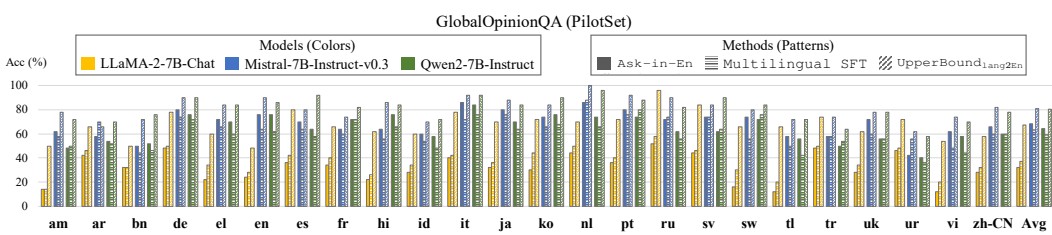

Figure 9: Main results on culture-aware task. Colors distinguish different LLMs, while patterns indicate applied methods. `Multilingual SFT` represents the results after multilingual supervised fine-tuning, while `UpperBound`$_{lang2En}$ represents $\mathcal{X}$Transplant from non-English language to English.

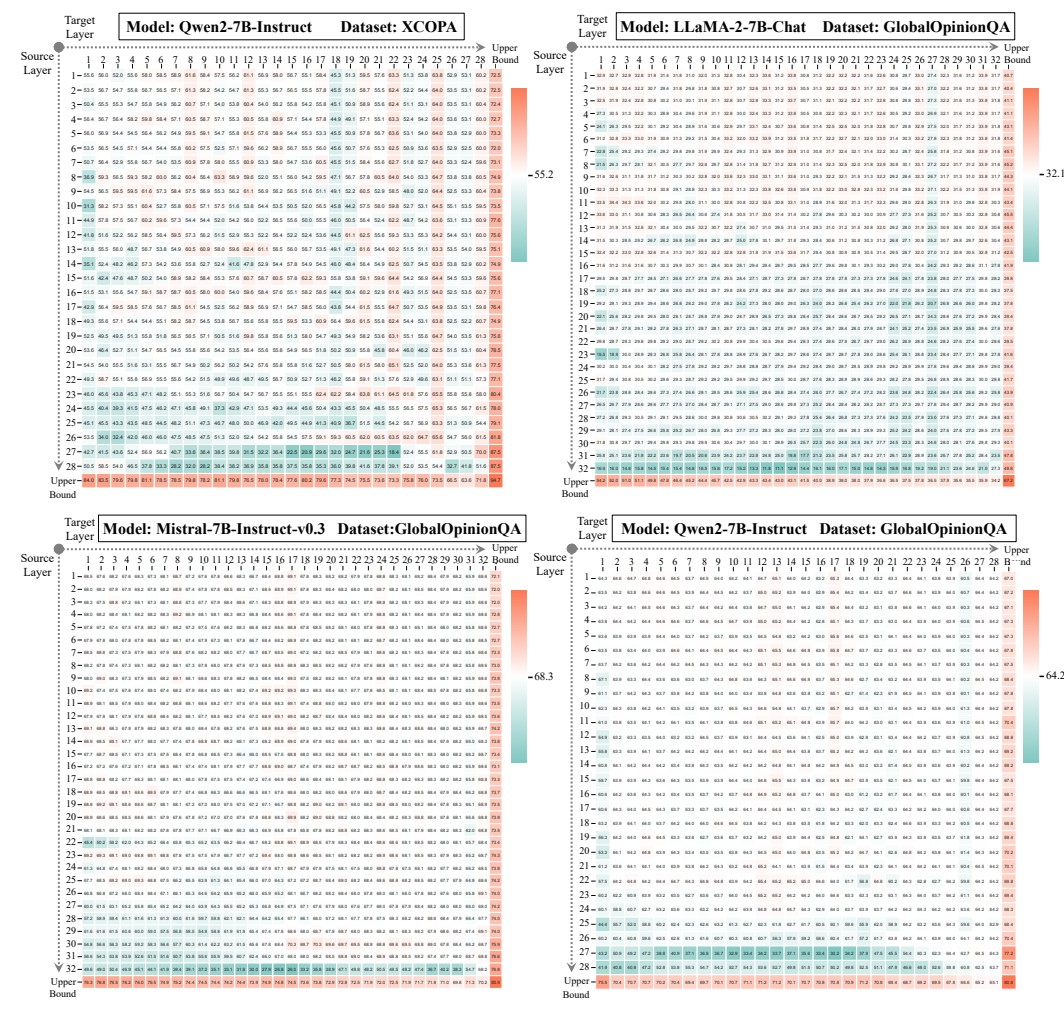

Figure 10: Supplementary layer-specific effectiveness results (part 2).

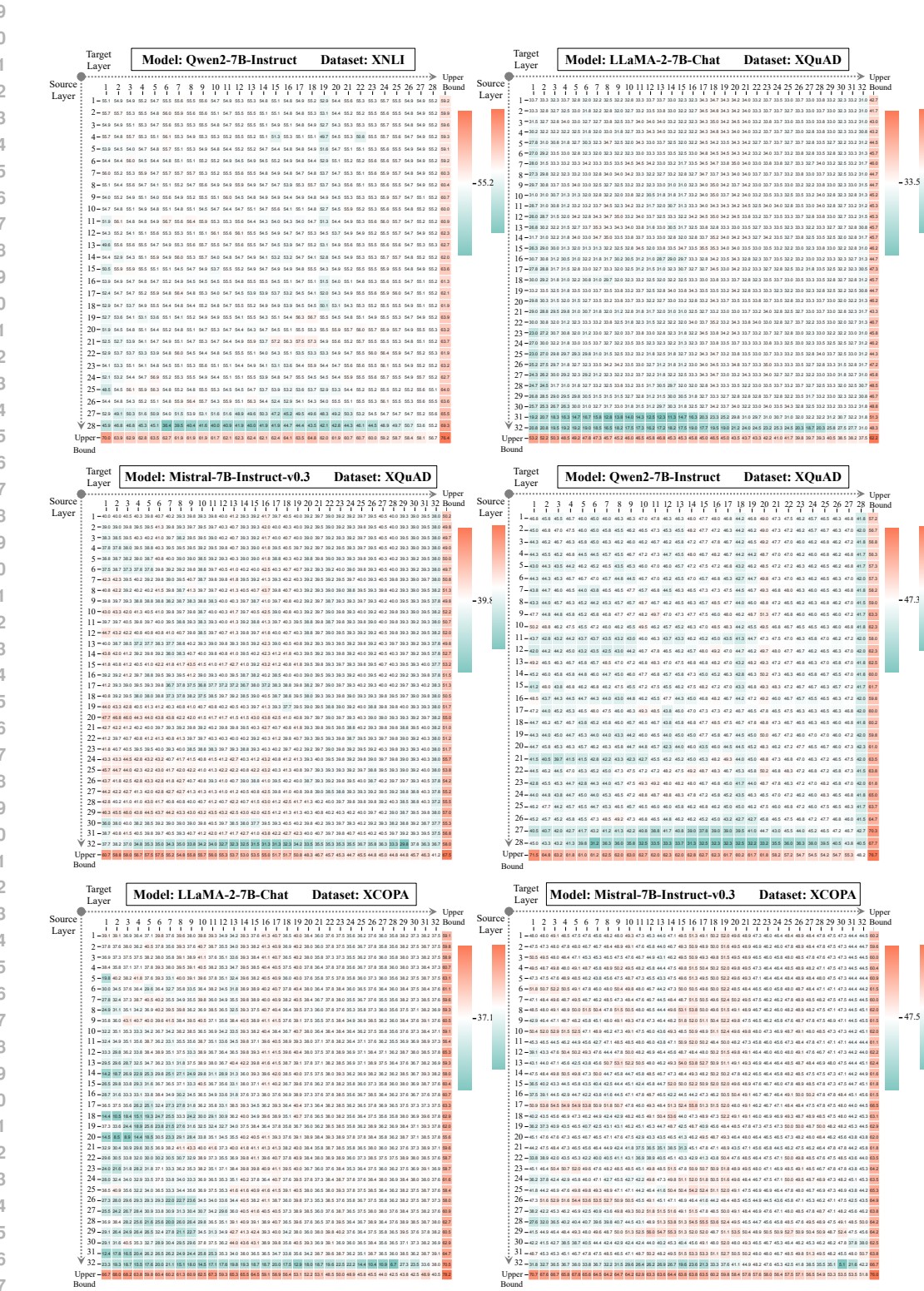

Figure 11: Supplementary layer-specific effectiveness results (part 1).

