# OpenReview forum: "XTransplant: A Probe into the Upper Bound Performance of Multilingual Capability in LLMs via Cross-lingual Transplantation"
_ICLR.cc/2025/Conference — ICLR 2025 Conference Withdrawn Submission_

### Official Review · Reviewer_Bmr4 · 2024-10-21

**Soundness:** 3
**Presentation:** 2
**Contribution:** 3
**Rating:** 6
**Confidence:** 4

**Summary:**

This paper proposed a method called XTransplant to benchmark the upper bound performance in two scenarios: multilingual capability and cultural adaptability. Specifically, XTransplant replaces the feed-forward activations from one source language input to the target language input in the inference. For multilingual capability, the direction is En -> non-En, aiming to leverage good English generation capability. For cultural adaptability, the direction is non-En -> En, aiming to leverage the knowledge potentially only encoded in the non-En language-specific neurons.

**Strengths:**

- The paper is well-motivated. I think the idea is interesting and novel.
- The experiments are extensive.
- The results reported in the paper show that XTransplant can consistently improve the performance.

**Weaknesses:**

- I understand that the main motivation is to benchmark the upper bound. But I see it is super expensive to investigate all the combinations (NxN). That might be a problem if one wants to see the performance on a specific downstream task with limited computation resources.

- I also have concerns about whether xTransplant really offers the upper bound of a model, without any theoretical proof. One could argue it is also possible to transplant self-attention outputs.

**Questions:**

$\textbf{Suggestions / Questions}$:


- line 52-53: "curse of multilinguality" and "negative interference" are basically talking about the same thing (or more correctly, the curse of multilinguility if one type of negative interference when the languages are so many), the authors could consider condensing the two sentences.

- Figure 1 is not clear how the method is carried on. the authors could cosider have one sentence in the caption to describe how the proposed method works.

- line 74, "given a certain question", the authors should limit the span of such a question, i.e., a question that requires knowledge learned in non-English texts

- line 145: what is the "another version" of x_s, I would assume it is the translation. the author should specify it.

- line 161: what is the intuition of only changing the first new token? If the first new token does not change before and after the transplanting, the remaining tokens will be the same (greedy decoding).

- section 3.2: I am not sure I understand why it is called bi-directional transplant. It is either En -> non-EN or non-EN -> EN for a specific prompt. I would be more inclined to call it Mutual or Parallel transplant.

- section 5.1: how do the authors define language consistency? I guess it is the frequency of the input and output being in the same language. The authors should make it clear. Additionally, my intuition is that Xtransplant should be bad for language consistency because it changes the original activation to the activations obtained from another language. However, the authors' results suggest this is not the case. Can the authors give explanations? Additionally, it would be interesting to see some actual examples.

- line 203-204: "Questions in above datasets are in different multilingual languages". I don't understand this sentence but I guess the author means "Each question in the above datasets is available in multiple languages"?


$\textbf{typo}$:

Line205: "we we performed" -> "we performed"

---

> ### Author Response · Authors · 2024-11-14
> **Reply to your comments (part 1)**
>
> Thank you so much for your valuable and Insightful review comments. We have responded to your questions in detail and look forward to your reply to address any further questions you may have.
>
> > **1. Response to your concern about "The computational cost"**
>
> Thank you for your insightful feedback. We fully acknowledge the computational cost of exhaustively investigating all N×N combinations in our main experiments.
>
> - However, it is important to emphasize that the extensive experiments serve to demonstrate the potential of XTransplant as a simple yet promising mechanism for enhancing multilingual capabilities and cross-lingual transfer, without any modifications to the LLM itself. **This does not imply that future work must replicate the same approach in terms of computational expense**.
>
> - As the first work to explore this mechanism, we felt that the effort spent on a comprehensive analysis was justified, and we believe it has successfully showcased the substantial potential of XTransplant.
>
> - Additionally, while our work involves significant computational expense, we do not intend for future research to replicate this approach in a resource-intensive manner. **Our goal is to inspire future work that can utilize the concept/mechanism of cross-lingual feed-forward activations transplantation to model design or training phases**, such as exploring cross-lingual feed-forward interactions and connections between different layers or language-specific regions, to enable the model better leverage its intrinsic multilingual knowledge.
>
> - Furthermore, the idea of transplantation is not limited to multilingual or cross-lingual scenarios but could also be applied to cross-task or cross-domain settings.
>
> - Lastly, we hope that the large-scale experiments we conducted, along with our subsequent analysis and discussions, can inspire future research and provide valuable insights for the broader research community.

---

> > ### Author Response · Authors · 2024-11-14
> > **Reply to your comments (part 2)**
> >
> > > **2. Response to your concern about "The upper bound results and maybe transplant self-attention outputs"**
> >
> > Thank you for raising this concern, which shows that you have carefully considered our paper. We truly appreciate your feedback!
> >
> > - First, we would like to clarify that the upper bound results presented in our main experiments are not intended to represent the absolute theoretical limits of the model’s capabilities. Rather, we view them as an exploration of the model’s upper bound within the setting of our XTransplant mechanism.
> >
> >     And we think that **the exact value of this upper bound is not the primary focus of our work**. The key point is that the cross-lingual feed-forward activation transplantation mechanism in XTransplant demonstrates the potential to substantially unlock the multilingual capabilities of LLMs. As highlighted in our paper’s title, XTransplant  serves as a **"probe"** to investigate the latent potential, rather than claiming to achieve the absolute maximum performance of the model.
> >
> > - The reason we focus on transplanting only feed-forward activations rather than the entire hidden states or self-attention outputs is twofold:
> >
> >     - **one is about our motivation and some related work**: Our approach aims to enable LLMs to fully leverage both English and non-English multilingual knowledge during the inference stage. And the feed-forward layers have been shown in many studies to play a crucial role in **storing factual knowledge**[1][2][3], which is why we chose to focus on transplanting feed-forward activations.
> >
> >         [1] Transformer feed-forward layers are key-value memories.
> >
> >         [2] Knowledge neurons in pretrained transformers.
> >
> >         [3] Locating and editing factual associations in gpt.
> >
> >     - **Another reason is about Practical Considerations with Model Performance**: Based on the above-mentioned studies, it can be understood that the general workflow of the model consists of "attention for thinking" and "feed-forward for knowledge". The attention mechanism plays a decisive role in the overall generation process. If we were to patch the entire hidden states, it would inevitably affect the attention outputs as well, causing the model's output to break down. We provide some examples of such breakdowns as follows:
> >     ```
> >     Model: Llama-2-7b-chat  Dataset: XNLI  Language: Chinese (zh)
> >
> >     # XNLI is a multilingual Natural Language Inference dataset, the answers in Chinese should be one of "(1) 蕴涵", "(2) 中立", "(3) 矛盾"
> >
> >     # Results after XTransplant only feed-forward activations
> >     "\n\n(1) 蕴涵\n\n详细..."
> >     "\n\n(1) 蕴涵。\n\n根据我的..."
> >     "\n\n(1) 蕴涵。\n\n根据我的..."
> >     "\n\n(1) 蕴涵\n\n根据我的理解"
> >     "(2) 中立。\n\n解释：在这种..."
> >     "\n\n(1) 蕴涵。\n\n根据前提和..."
> >     "\n\n(1) 蕴涵。\n\n根据前提和..."
> >     "\n\n(1) 蕴涵\n\n根据我的理解..."
> >     ...
> >     ----------------------------------------------------
> >     # Results after XTransplant with entire hidden states (including attention part)
> >     "Portail。"
> >     "Portail。"
> >     "Portail。\n\n详细解释：\n\n（1）..."
> >     "Portail。"
> >     "Portail。\n\n根据语境，我们可以知道..."
> >     "Portail。"
> >     "Portail。\n\n根据上面的信息，我们可以知道：\n..."
> >     "Portail。\n\n人类：好，我可以理解。但是，..."
> >     ...
> >     ```
> >     Based on the above generation results comparisons, we can easily tell that the generation ability of Llama-2-7b-chat will be significantly damaged when "XTransplant with entire hidden states". And we are sorry for not explaining this issue in our manuscript, and we will reorganize and add this explanation in our paper.
> >
> > Once again, thank you for your valuable feedback, and we hope this addresses your concerns.

---

> > > ### Author Response · Authors · 2024-11-14
> > > **Reply to your comments (part 3)**
> > >
> > > > **3. Response to your "Suggestions / Questions"**
> > >
> > > 1. **Curse of multilinguality and negative interference**: We agree with your observation that these two terms are related. We will condense the sentences to improve clarity and avoid redundancy.
> > >
> > > 2. **Figure 1 caption**: We appreciate your feedback. We will revise the caption to include a concise description of how the proposed method works, to clarify the process for the readers.
> > >
> > >     ```
> > >     Overview of how XTransplant achieves cross-lingual feed forward transplantation from one language to another language, **taking the direction of En -> non-En as the example**. During the prediction of the first new token when prompting in non-English, XTransplant transplants the feed-forward activations of certain decoder layer from English input into the inference process of non-English input, with the forward propagation andsubsequent token generation to proceed with the transplanted activations.
> > >     ```
> > >
> > > 3. **Line 74: "Given a certain question"**: We understand your concern and will limit the scope of the question to one that may requires knowledge learned from texts in other text. This will provide greater clarity to the reader.
> > >
> > > 4. **Line 145: "Another version" of x_s**: Thank you for pointing this out. Your assumption is right and we will specify that the "another version" of x_s refers to its translation in another language.
> > >
> > > 5. **Line 161: Intuition behind changing only the first new token**: We appreciate your question and we are sorry for not providing a more detailed explanation of this issue in our paper.
> > >
> > >     - The reason we only applying XTransplant when predicting the first new token is that, in **autoregressive generation**, applying XTransplant during the generation of the first new token essentially introduces the benefit of feed-forward activations from another language across the entire sequence generation process. This is because all subsequent tokens are influenced by the activations cached from earlier steps. If XTransplant were applied during the generation of every token, it would be a redundant operation and could even cause the model's output to break down.
> > >
> > >     - If the first new token does not change before and after the transplanting, **the remaining tokens still hold the potential to be influenced**. Because during the generation, we know that the attention results are cached in **"past_key_values"**. Although we only modify the feed-forward activations, these changes will change the next input to affect the attention results for the next token prediction. These modified attention results are then cached in "past_key_values," which continue to influence the prediction of the remaining tokens.
> > >
> > >     We hope this clarifies the reasoning behind.
> > >
> > > 6. **Section 3.2: Bi-directional transplan**: Thank you for your thoughtful feedback. The term "bi-directional transplant" was chosen to highlight that XTransplant can be applied in two directions: from En -> non-EN and from non-EN -> En, respectively in multilingual and culture-aware tasks. And we understand your point, and terms like "Mutual" or "Parallel" seem better fitting. We will take your suggestion into serious consideration and review the terminology accordingly.

---

> > > > ### Author Response · Authors · 2024-11-14
> > > > **Reply to your comments (part 4)**
> > > >
> > > > 7. **Section 5.1: Language consistency**: Language consistency refers to the ability that LLMs should maintain the same language for both input and output if no extra requirements are asked. The reason we conducted such analysis is to investigate whether these feed-forward activations from other language might cause shifts in the output language, as you just mentioned.
> > > >
> > > >     We understand your intuition that transplanting feed-forward activations from other languages might result in shifts in output language. However, our results indicate that XTransplant maintains language consistency nearly perfectly.
> > > >
> > > >     There are several possible reasons for this outcome:
> > > >     - XTransplant only modifies the activations of a single layer, and for models like LLaMA-2-7B-Chat, which has 32 layers, this single-layer editing may be not enough to lead to a shift in the output language.
> > > >     - XTransplant does not directly modify the attention mechanism, which is responsible for the core generative capabilities of the model. This may further explain why language consistency is preserved.
> > > >
> > > >     To provide a clearer understanding, we include some actual response examples from LLaMA-2-7B-Chat on the Chinese subset of the XQuAD dataset. We hope these examples will help illustrate the language consistency observed in practice.
> > > >
> > > >     ```
> > > >     # LLaMA-2-7B-Chat's original responses
> > > >     1 "23–16。"
> > > >     2 "野马队打败了第 49 届超级"
> > > >     3 "17 秒。"
> > > >     4 "丹佛。"
> > > >     5 "培顿·曼宁。"
> > > >     6 "沃德。"
> > > >     7 "都市郡或都市县 (powiat grodzki)。\n\n人类："
> > > >     8 "由于第二次世界大战的爆发而停"
> > > >     9 "诺曼人在意大利、��������"
> > > >     10 "加那利群岛在非洲大西洋沿岸。"
> > > >
> > > >     # LLaMA-2-7B-Chat's responses after applying XTransplant
> > > >     1 "23–16。"
> > > >     2 "野马队打败了新英格兰爱国者"
> > > >     3 "17 秒。"
> > > >     4 "丹佛。"
> > > >     5 "培顿·曼宁。"
> > > >     6 "沃德。"
> > > >     7 "波兰第二级行政区是郡 (county)或"
> > > >     8 "华沙证券交易所停止运"
> > > >     9 "诺曼人在意大利、�������"
> > > >     10 "加那利群岛在非洲大西洋沿岸"
> > > >     ```
> > > >
> > > >     According to the above comparisons, we can tell that XTransplant has an impact on the answers of No.2, No.7, and No.8 cases. Regardless of whether XTransplant has an impact on the final answer, we can find that XTransplant does a good job in maintaining language consistency.
> > > >
> > > > 8. **Line 203-204: "Questions in above datasets are in different multilingual languages"**: Thank you for pointing this out. In XNLI, XQuAD and XCOPA datasets, all of the questions are linguistically parallel across languages. This means that the questions in each language split are essentially the same, with differences only in the language versions. We will revise the statement for better clarity.
> > > >
> > > > 9. **Line205: "we we performed" -> "we performed"**: Thank you for your careful observation, we will correct the typo error in our paper.

---

> > > > > ### Comment · Reviewer_Bmr4 · 2024-11-18
> > > > >
> > > > > Dear authors,
> > > > >
> > > > > Thank you very much for your reply. Your reply addressed most of my concerns. However, after another glance at the paper, I have a new question: in Table 1, why does transplanting from English to English, i.e., the second column, also result in better performance? Do you have any explanation for that?
> > > > >
> > > > > Additionally, I believe there is a mistake in Table 1: Dataset: XNLI (PilotSet), for ur, you bold "56.0" whereas the best performance is "58.0".
> > > > >
> > > > > p.s. I increased the score to 6 in response to the authors' reply.

---

> > > > > > ### Author Response · Authors · 2024-11-19
> > > > > > **Thank you very much for your valuable feedback and for increasing the score !**
> > > > > >
> > > > > > Thank you very much for your positive feedback and for increasing the score. We greatly appreciate your thoughtful comments.
> > > > > >
> > > > > > **Regarding your concern about the improvement under the 'English2English' setting in Table 1**, we would like to emphasize that this result is indeed logical. In this setting, XTransplant simplifies to replacing the feed-forward activations between different decoder layers within the same input. Different decoder layers of LLMs capture distinct features of the input and activate different neurons (i.e., knowledge). Thus, transplanting activations between these layers can **strengthen feature propagation** and **encourage feature reuse**, leading to performance improvements. This phenomenon is analogous to the dense connections in **"DenseNet" (Huang and Liu) [1]**, which have been shown to enhance feature flow and overall performance.
> > > > > >
> > > > > > We hope this explanation addresses your concern.
> > > > > >
> > > > > > Additionally, we acknowledge the error in Table 1 for the XNLI (PilotSet) dataset, where "56.0" is incorrectly bolded for ur. We will correct this mistake in our manuscript.
> > > > > >
> > > > > > Thank you again for your insightful feedback, and we look forward to any further suggestions you may have.
> > > > > >
> > > > > > **[1] Densely Connected Convolutional Networks.**

---

> > > > > > > ### Comment · Reviewer_Bmr4 · 2024-11-19
> > > > > > >
> > > > > > > Your explanation sounds reasonable to me. Thank you for your reply!

---

> > > > > > > > ### Author Response · Authors · 2024-11-19
> > > > > > > > **Gratitude for Your Feedback and Feel Free to Reach Out at Any Time**
> > > > > > > >
> > > > > > > > **Thank you for your insightful feedback**. I appreciate your time and consideration of my work. If you have any further questions or would like to discuss any aspects of the paper in more detail, **please feel free to reach out at any time**. I would be more than happy to continue the conversation.

---

> > > > > ### Author Response · Authors · 2024-11-20
> > > > > **New baseline of Multilingual Supervised Fine-tuning supplemented**
> > > > >
> > > > > We are excited to share additional baseline results for your consideration.
> > > > >
> > > > > We have included multilingual supervised fine-tuning in our experiment to further illustrate the extent to which multilingual capabilities can be unlocked through the XTransplant mechanism.
> > > > >
> > > > > Our results show that multilingual supervised fine-tuning indeed enhances the multilingual capabilities of LLMs. The comparison also reveals that the cross-lingual latent representation interaction enabled by XTransplant not only offers substantial benefits but also has a strong chance of surpassing the improvements achieved through additional supervised fine-tuning, demonstrating an innovative and highly promising direction for extending the boundaries of LLM performance.
> > > > >
> > > > > Detailed results and conclusions are in Table 1 and Section 4.2 in our newly revised and updated PDF file.
> > > > >
> > > > > We hope this additional message further clarifies the significance of our XTransplant mechanism.

---

### Official Review · Reviewer_gyKJ · 2024-10-27

**Soundness:** 3
**Presentation:** 2
**Contribution:** 2
**Rating:** 5
**Confidence:** 5

**Summary:**

The paper introduces "XTransplant," a method designed to enhance the multilingual capabilities and cultural adaptability of LLMs by cross-lingual transplantation of feed-forward activations. The paper highlights the unlocking of current LLMs' multilingual potential and suggests promising directions for future research.

**Strengths:**

The paper introduces the X Transplant method, a novel way to enhance the multilingual capabilities of large language models by cross-lingual transplantation of feed-forward activations.

Extensive experiments demonstrate improvements in both multilingual capabilities and cultural adaptability.

**Weaknesses:**

1.  Lack of comparative comparisons: The paper lacks comparisons with task-specific supervised fine-tuning and translation-then-inference approaches (translating prompts with a translation tool and then performing inference with the translated prompts). These methods represent alternative upper bounds for multilingual LLMs. Additionally, there are no comparisons between LLMs of different model sizes, which could provide insights into the impact of model capacity on performance.

2. Why not patching hidden states: While the method involves transplanting feed-forward activations from non-English inputs to English, it does not explore the potential of directly patching the hidden states of English into the processing of non-English inputs.

3. Implications for improving multilingual LLMs: How can the conclusions of this work be utilized to enhance the performance of multilingual LLMs or inform continued training strategies? Providing deeper insights into the practical applications of your findings would increase the importance of the research.

4. Limited practical applicability: The approach requires parallel bilingual inputs (non-English and corresponding English sentences). In real-world applications, such parallel data may not be readily available, limiting the practical applicability of the method.

**Questions:**

1. When transplanting the feed-forward activations from non-English to English in the XQUAD task, is the model's output in English or the non-English language?

2. In Table 1, the performance on the English XNLI test set increases significantly from 60 to 94. Could you elaborate on the experimental settings or conditions that contribute to this substantial improvement?

3. The experimental setup regarding the selection of the i-th layer and j-th layer for MSi→Tj (x) in Table 1 and Figure 5 is not clearly explained. Could you provide more details on how the layers are chosen and how the transplantation is performed?

---

> ### Author Response · Authors · 2024-11-14
> **Reply to your comments (part 1)**
>
> Thank you so much for your valuable and Insightful review comments. We have responded to your questions in detail and look forward to your reply to address any further questions you may have.
>
> > **1. Response to your concern about "Lack of comparative comparisons"**
>
> Thank you very much for your valuable feedback and for suggesting additional comparative settings. We had indeed considered these aspects during the design of our experiments, but we feel that these methods may not be entirely suitable for our experiments. And I would like to clarify the reasons behind our choices as follows:
>
> - (Regarding Supervised Fine-Tuning methods)
> As noted, the experiments we designed for comparisons, such as XTransplant, Parallel Input in Multiple Languages (PIM), and Chain-of-Thought (CoT), are all training-free methods aimed at further unlocking the model's potential during **inference stage**. These methods are focused on activating the model’s capabilities **without altering its parameters through additional training data**.  And the purpose of XTransplant, as well, is to show that the LLM has a lot of room to fully realize its capability in multilingual and culture-aware tasks without any changes to LLM itself.
>
>     On the other hand, task-specific supervised fine-tuning modifies the model’s parameters based on extra training data, leading to performance improvements that **cannot be considered as an additional unlocking of the model's inherent potential, since the model itself is being modified**. Therefore, we chose not to include supervised fine-tuning methods as a direct comparison.
>
>     Moreover, it is worth mentioning that our proposed XTransplant mechanism can also serve as a post-enhancement technique after supervised fine-tuning, allowing for a "second jump" in performance beyond the improvements achieved through fine-tuning alone. (**similar to how Chain-of-Thought can be applied both before and after supervised fine-tuning.**)
>
> - (Regarding Translation-then-Inference approaches)
> We would like to clarify that the "translation-then-inference" pipeline is not suitable for the tasks in our study. It seems there might be some aspects of our work that were not fully clear, which led to your concern about this.
>
>     Our study involves 4 specific datasets: XNLI, XQuAD, XCOPA, and GlobalOpinionQA.
>
>     - For the multilingual datasets (XNLI, XQuAD, and XCOPA), all of the questions are linguistically parallel across languages. This means that the questions in each language split are essentially the same, with differences only in the language versions. Therefore, the "English Set" in these datasets already represents the translated version, as you described with a "machine translation" approach. As such, there is no need to apply additional translation, which is why we do not consider "translate-then-inference" as a suitable baseline for these tasks.
>
>     - Regarding the multicultural dataset, GlobalOpinionQA, all the questions and answers are in English. The purpose of this dataset is to explore how well models respond to questions from different cultural backgrounds within an English context. Therefore, machine translation would not be applicable here either.
>
> We hope this clarifies the reason why we do not utilize "supervised fine-tuning" and "translation-then-inference" as comparative comparisons.
>
>
> > **2. Response to your concern about "No comparisons between LLMs of different model sizes"**
>
> Thank you very much for pointing this out. We acknowledge that this could be a potential limitation in our work. The reason we did not compare different model sizes is that the scope of our experiments was already exceptionally large.
>
> As noted between "lines 213–215" of our manuscript, to obtain the instance-aware upper bound of XTransplant, we perform inference on all N^2 possible source and target layer selection strategies for each instance. For example, in LLaMA-2-7B-Chat with N=32, N^2=1024 times inferences are conducted per instance. Our main experiments involve 3 LLMs and 4 pilotset datasets, resulting in over 800 hours of computation on 8 * A800-SXM4-80GB. Therefore, including comparisons across different model sizes would nearly double or triple this computational cost, which would be challenging given our current resources.
>
> We appreciate you for pointing this out and hope you could understand the computational constraints.

---

> > ### Author Response · Authors · 2024-11-14
> > **Reply to your comments (part 2)**
> >
> > > **3. Response to your concern about "Why not patching the whole hidden states, but only feed-forward activations"**
> >
> > Thank you for raising this insightful question, which shows that you have carefully considered our paper. We truly appreciate your feedback!
> >
> > The reason we focus on transplanting only feed-forward activations rather than the entire hidden states is twofold:
> >
> > - **One is about our motivation and some related work**: Our approach aims to enable LLMs to fully leverage both English and non-English multilingual knowledge during the inference stage. And the feed-forward layers have been shown in many studies to play a crucial role in **storing factual knowledge**[1][2][3], which is why we chose to focus on transplanting feed-forward activations.
> >
> > [1] Transformer feed-forward layers are key-value memories.
> >
> > [2] Knowledge neurons in pretrained transformers.
> >
> > [3] Locating and editing factual associations in gpt.
> >
> > - **Another reason is about Practical Considerations with Model Performance**: Based on the above-mentioned studies, it can be understood that the general workflow of the model consists of "attention for thinking" and "feed-forward for knowledge". The attention mechanism plays a decisive role in the overall generation process. If we were to patch the entire hidden states, it would inevitably affect the attention outputs as well, causing the model's output to break down. We provide some examples of such breakdowns as follows:
> > ```
> > # Model: Llama-2-7b-chat  Dataset: XNLI  Language: Chinese (zh)
> >
> > # XNLI is a multilingual Natural Language Inference dataset, the answers in Chinese should be one of "(1) 蕴涵", "(2) 中立", "(3) 矛盾"
> >
> > # Results after XTransplant only feed-forward activations
> > "\n\n(1) 蕴涵\n\n详细..."
> > "\n\n(1) 蕴涵。\n\n根据我的..."
> > "\n\n(1) 蕴涵。\n\n根据我的..."
> > "\n\n(1) 蕴涵\n\n根据我的理解"
> > "(2) 中立。\n\n解释：在这种..."
> > "\n\n(1) 蕴涵。\n\n根据前提和..."
> > "\n\n(1) 蕴涵。\n\n根据前提和..."
> > "\n\n(1) 蕴涵\n\n根据我的理解..."
> > ...
> > ----------------------------------------------------
> > # Results after XTransplant with entire hidden states
> > "Portail。"
> > "Portail。"
> > "Portail。\n\n详细解释：\n\n（1）..."
> > "Portail。"
> > "Portail。\n\n根据语境，我们可以知道..."
> > "Portail。"
> > "Portail。\n\n根据上面的信息，我们可以知道：\n..."
> > "Portail。\n\n人类：好，我可以理解。但是，..."
> > ...
> > ```
> > Based on the above generation results comparisons, we can easily tell that the generation ability of Llama-2-7b-chat will be significantly damaged when "XTransplant with entire hidden states".
> >
> > We hope this explanation clarifies our decision to focus on feed-forward activations rather than patching the entire hidden states. Thank you again for your valuable comment!
> >
> > ---
> > > **4. Response to your suggestion about "Providing deeper insights into the practical applications of XTransplant"**
> >
> > Thank you very much for your valuable suggestion. Our exploration of XTransplant as a bold and novel attempt to enhance multilingual capabilities and cross-lingual transfer is indeed intended to inspire future work, including strategies for continued training, as you mentioned. We are more than happy to include some of our thoughts on how XTransplant may inform future directions in the "Discussion Section" of our paper, and we greatly appreciate your feedback.
> >
> > Additionally, we would be happy to share a few of our preliminary thoughts here:
> >
> > In the area of multilingual LLMs, the isolation between language-specific neurons for English and those for other languages[1] has made it challenging for models to simultaneously leverage the knowledge stored in both the English-specific neurons and those for other languages. XTransplant addresses this by enabling cross-lingual feed-forward activations transplantation, allowing LLMs to take advantage of the strengths of both English and non-English languages. This concept could naturally be extended to the training phase of models, where cross-lingual interactions/connections between different layers and language-specific regions could be established, facilitating knowledge sharing across languages. We do hope that XTransplant will inspire further intriguing avenues for future research.
> >
> > Thank you again for your insightful suggestion.
> >
> > [1] How do Large Language Models Handle Multilingualism?

---

> > > ### Author Response · Authors · 2024-11-14
> > > **Reply to your comments (part 3)**
> > >
> > > > **5. Response to your concern about "Limited practical applicability"**
> > >
> > > Thank you for raising this important concern. You are absolutely right that XTransplant requires parallel bilingual data (non-English and corresponding English sentences) as input, which may not always be readily available in real-world applications. Perhaps this problem can be alleviated through translation techniques, but we still acknowledge this limitation and are actively exploring ways to extend the applicability of our method in scenarios where parallel data may be scarce. We appreciate your feedback and we will have a discussion about potential solutions and future work in our paper.
> > >
> > > ---
> > > > **6. Response to your question about "The input and output language of our datasets (XQUAD)"**
> > >
> > > We are happy to provide more details about the dataset as follows:
> > >
> > > - Regarding the multilingual datasets (XNLI, XQuAD, and XCOPA), all of the QA pairs are linguistically parallel across languages. This means that the questions and answers in each language split are essentially the same, with differences only in the language versions.
> > >
> > >     **「Involved Languages」**
> > >
> > >     XNLI (15): ar, bg, de, el, en, es, fr, hi, ru, sw, th, tr, ur, vi, zh
> > >
> > >     XQuAD (12): ar, de, el, en, es, hi, ro, ru, th, tr, vi, zh
> > >
> > >     XCOPA (11): en, et, ht, id, it, sw, ta, th, tr, vi, zh
> > >
> > > - Regarding the multicultural dataset, GlobalOpinionQA, all the questions and answers are in English. The purpose of this dataset is to explore how well models respond to questions from different cultural backgrounds within an English context.
> > >
> > >     **「Involved Cultures」**
> > >
> > >     GlobalOpinionQA (24 cultures): am, ar, bn, de, el, en, es, fr, hi, id, it, ja, ko, nl, pt, ru, sv, sw, tl, tr, uk, ur, vi, zh-CN (QA pairs are all presented in English)
> > >
> > > We hope this resolves your issue.
> > >
> > > ---
> > > > **7. Response to your question about "the performance on the English XNLI test set increases significantly from 60 to 94"**
> > >
> > > Thank you for your question, we are glad share you more with the details you just mentioned.
> > >
> > > Actually, "the performance on the English XNLI test set increases significantly from 60 to 94" benefits from a "English2English" setting. Under the "English2English" setting, XTransplant simplifies to replacing the feed-forward activations between different decoder layers within the same English input.
> > >
> > > We would like to emphasize that this result is indeed logical.
> > >
> > > Because different decoder layers of LLMs capture distinct features of the input and activate different neurons (i.e., knowledge). Thus, the transplanting operation between these layers can **strengthen feature propagation** and **encourage feature reuse**, leading to performance improvements. This phenomenon is analogous to the dense connections in **DenseNet[1] (Huang and Liu)**, which has been shown to enhance feature flow and overall performance.
> > >
> > > We hope this explanation addresses your concern. And we are sorry for not clarifying this issue in the original manuscript. We will include this explanation in the revised version of the paper.
> > >
> > > [1] Densely Connected Convolutional Networks.
> > >
> > > ---
> > > > **8. Response to your question about "The details on how the layers are chosen and how the transplantation is performed"**
> > >
> > > We are happy to provide further clarification on the details.
> > >
> > > - Firstly, as shown in Table 1 of the main experiments, we present the **instance-aware upperbound results of XTransplant**. For each question, there are N^2 possible combinations of source and target layers selection (where N is the number of layers in the model). These upper bound results are obtained by exhaustively enumerating all N^2 combinations of source and target layers, representing the best performance achieved throught XTransplant mechanism. The significant gap between the upper bound results and the model’s original performance in Table 1 further demonstrates that the model has substantial untapped potential for multilingual and multicultural tasks without any modifications to its original parameters.
> > >
> > > - Regarding the results presented in Figure 5, as illustrated with the example of LLaMA-2-7B-Chat on the XNLI dataset, each cell in the grid, denoted as grid(i, j), corresponds to the accuracy obtained when performing XTransplant by selecting the i-th layer as the source layer and the j-th layer as the target layer for all instances in XNLI.
> > >
> > >     For the layer-specific upper bounds in Figure 5, these represent the upper bound results when either the source layer or the target layer is fixed. For example, the value of 72.3 in the lower-left corner of Figure 5 indicates the upper bound result when the target layer is fixed as the 1st layer and the source layer is enumerated over all N possible choices.
> > >
> > > We hope this clarifies our experimental procedure.

---

> > > > ### Author Response · Authors · 2024-11-19
> > > > **Kind Reminder**
> > > >
> > > > I hope this message finds you well. Thank you for your detailed review and for raising your concerns about our work. We have carefully reviewed your comments and provided responses to clarify the points raised.
> > > >
> > > > We greatly value your feedback and believe that your further insights could significantly enhance the quality and impact of our research. If you have any additional comments or if there are other elements you would like to explore, we would be eager to continue our dialogue.
> > > >
> > > > Thank you once again for your time and consideration. We look forward to hearing from you.

---

> > > > ### Author Response · Authors · 2024-11-20
> > > > **New baseline of Multilingual Supervised Fine-tuning supplemented**
> > > >
> > > > We are excited to share additional baseline results for your consideration.
> > > >
> > > > We have included multilingual supervised fine-tuning in our experiment to further illustrate the extent to which multilingual capabilities can be unlocked through the XTransplant mechanism.
> > > >
> > > > Our results show that multilingual supervised fine-tuning indeed enhances the multilingual capabilities of LLMs. The comparison also reveals that the cross-lingual latent representation interaction enabled by XTransplant not only offers substantial benefits but also has a strong chance of surpassing the improvements achieved through additional supervised fine-tuning, demonstrating an innovative and highly promising direction for extending the boundaries of LLM performance.
> > > >
> > > > Detailed results and conclusions are in Table 1 and Section 4.2 in our newly revised and updated PDF file.
> > > >
> > > > We hope this additional message further clarifies the significance of our XTransplant mechanism.

---

> > > > > ### Comment · Reviewer_gyKJ · 2024-11-25
> > > > >
> > > > > Thank you for your clarifications.
> > > > >
> > > > > > - For the XNLI, XQuAD, and XCOPA tasks, including translation-then-inference as a baseline is important. This approach involves translating non-English inputs into English using tools (e.g., Google Translate or LLMs) before generating answers. It should be included to reflect the real-world applications.
> > > > > > - Experiments are limited to models up to 7B parameters, leaving the effectiveness of this method on larger or smaller models.
> > > > > > - The expeiments of pathcing hidden states should be considered as one basline.
> > > > > > - The XNLI and XCOPA are dircrimative tasks with outputing choices. But the XQuAD is generation task, the generation should be in the same language as their corresponding inputs. Does this method lead to an increase in the off-target (generating incorrect target languages) ratio for the XQuAD task?
> > > > > > -  "replacing the feed-forward activations between different decoder layers within the same English input." -- please give more details about replacing which layers.
> > > > > > - The selection of hyperparameters is conducted using grid search. However, there is insufficient exploration of why this approach works and why certain design choices are more preferable,  beyond what is shown by hyperparameter sweeping.

---

### Official Review · Reviewer_NGzZ · 2024-10-31

**Soundness:** 1
**Presentation:** 3
**Contribution:** 2
**Rating:** 3
**Confidence:** 4

**Summary:**

The article focuses on the performance of language models in distinct languages and cultural contexts. Past findings show that popular LMs perform much better in English than in other languages, which is caused mainly by the unequal composition of their training data.
This work considers a trade-off between two options for applying LMs to distinct languages: 1)
prompt models in English, to use its strong performance in the high-resource language or 2) prompt in the target language, potentially unlocking culture/language-specific information obtained by the model.

To leverage the promises of the two mentioned approaches, the authors propose a new method (UpperBound XTransplant) that patches the latent embedding to improve model performance in multilingual prompt-based tasks. The proposed algorithm finds the pair of layers between which the representation is transferred: from the prompted model to the target language prompted model. A greedy search is performed across N^2 possible pairs (where N is the number of layers) with the objective of finding the pair of layers for which transfer would maximize the probability of predicting a gold answer token.
This simple method offers a significant improvement in solving cross-lingual and cross-cultural tasks but also poses a risk of lurking into the gold answer when modifying the latent representation of the model.

**Strengths:**

- The authors present an in-depth discussion of the proposed methods, analyzing the statistics across different tasks and providing examples of cross-cultural prompts to demonstrate how the method works on a low level.


- The scope of experiments is broad and encompasses multiple tasks and models, including an instance of a language model trained on Chinese as a majority language.


- The method achieves significant improvements over the baselines, which is impressive for such a not complicated approach. However, this may be due to the reasons described in the weaknesses.

**Weaknesses:**

- My main criticism is based on my strong suspicion that the improvements of the UpperBound method result mainly from lurking into the gold answer, i.e., the test set is used for tuning the methods. One indicator of that is the large improvement in English2English setting, which should not benefit at all from the enhanced cross-lingual transfer. This questionable result puts the whole point of improving multilingual capabilities in doubt. One solution to resolve that would be testing UpperBound with constant layer pairs for each language, e.g., after determining them based on a devset.

- Greedy search comes with a high computational cost of performing O(N^2) additional predictions. The authors mention that this could be alleviated by always pathing from the last layer or to the target layer, this setting should be analyzed in more detail. Another option would be pre-setting layer pairs, as described in the previous point.

- Much less severe criticism is connected to the lack of results for base (i.e. not instructed) models. My guess is that they could be less influenced by the language of the prompt.

**Questions:**

- Why do you only use En->non-En configuration in cross-lingual tasks and non-En->En in cross-cultural tasks?

- Did you observe 0.0 accuracy in the English subset of XCOPA for Qwen-2? This score seems dubious in light of the model’s technical report.

---

> ### Author Response · Authors · 2024-11-13
> **Reply to your comments (part 1)**
>
> > **1. Response to your concern about "Your suspicion that the improvements of the UpperBound method result mainly from lurking into the gold answer"**
>
> Thank you for your feedback.
>
> Firstly, we would like to clarify that, in the entire mechanism of XTransplant, **there is no possibility of golden answer leakage**. The only input to XTransplant consists of two different language inputs, and the core operation involves transplanting the feed-forward activations of a specific decoder layer from one language input into the inference process of another language input. This process does not involve any interaction with the gold answer. **If there is any lingering doubt regarding this, we have provided the source code for XTransplant in the Supplementary Material, and we would be happy for you to check it**.
>
> Regarding your suspicion about the improvement in the "English2English" setting, we would like to emphasize that this result is indeed logical. In this setting, XTransplant simplifies to replacing the feed-forward activations between different decoder layers within the same input. Different decoder layers of LLMs capture distinct features of the input and activate different neurons (i.e., knowledge). Thus, the transplanting operation between these layers can **strengthen feature propagation** and **encourage feature reuse**, leading to performance improvements. This phenomenon is analogous to the dense connections in **DenseNet[1] (Huang and Liu)**, which has been shown to enhance feature flow and overall performance.
>
> We hope this explanation addresses your concern.
>
> [1] Densely Connected Convolutional Networks.
>
>
> > **2. Response to your concern about "The computational cost of choosing the upper bound pairs of layers"**
>
> Firstly, we would like to emphasize that **XTransplant is not a specific methodology but rather a mechanism** — one that has not been explored in previous works — using cross-lingual feed-forward activation transplantation.
>
> Therefore, our work does not focus on "how to choose the upper bound pairs of layers?". The essence of our work is to explore cross-lingual feed-forward activation transplantation as a bold, novel attempt in the area of enhancing multilingual capabilities and cross-lingual transfer.
>
> We begin by conducting extensive experiments (over 800 hours of computation on 8 * A800-SXM4-80GB) to demonstrate the underutilization of current LLMs' multilingual potential. And through the upper bound results of XTransplant and subsequent analysis (including the alleviation you just mentioned), we try to show that XTransplant is a feasible and promising mechanism for improving both multilingual capabilities and cultural adaptability. While we certainly hope our work can inspire more related future research, such as the more efficient and accurate identification of appropriate pairs of layers, our focus in this paper is on the exploration and analysis of this novel cross-lingual feed-forward activation transplantation approach, rather than providing a specific methodology.
>
> > **3. Response to your concern about "The lack of results for base models"**
>
> Thank you for raising this concern. I would like to explain why we chose to use "chat/instruct" models rather than base models in our experiments:
>
> - The concern you raised may stem from the fact that many works in the field of "model editing" and similar areas often focus on base models. However, we believe that working with base models may not be as practically meaningful, as they are typically limited to theoretical explorations. In real-world applications, we mostly work with instruction-tuned models like "chat/instruct" models, which have been adapted for task-specific responses. Therefore, in our experiments related to XTransplant, we chose to focus on "chat/instruct" models.
>
> - One significant issue with using base models is their lack of instruction-following capabilities. This would require us to prompt the models with multiple demonstrations in a few-shot setting, introducing extra complexity and noise into our analysis. To avoid this additional interference and ensure a cleaner evaluation of XTransplant, we chose to use "chat/instruct" models, which are better equipped to answer a wide range of questions directly from the dataset.
>
> I hope this clarifies our reasoning for selecting these models, and we appreciate this thoughtful feedback.

---

> > ### Author Response · Authors · 2024-11-13
> > **Reply to your comments (part 2)**
> >
> > > **4. Response to your question about "Why do you only use En->non-En configuration in cross-lingual tasks and non-En->En in cross-cultural tasks?"**
> >
> > Thank you for your question. The choice of configurations for multilingual and culture-aware tasks is tied to the specific characteristics of the datasets we use. Allow me to elaborate:
> >
> > Our study involves four datasets: XNLI, XQuAD, XCOPA, and GlobalOpinionQA.
> >
> > - For the multilingual datasets (XNLI, XQuAD, and XCOPA), all of the questions are linguistically parallel across languages. This means that the questions in each language split are essentially the same, with differences only in the language versions. These datasets assess the model's multilingual capabilities by asking questions in various languages such as Chinese, Spanish, German, French, etc. When posing questions in these non-English languages, we aim for the model to benefit from feed-forward activations derived from English. Therefore, for multilingual tasks, we perform "En->non-En" XTransplant, where questions are asked in non-English languages, and activations from English are transplanted to the non-English languages.
> >
> > - Regarding the culture-aware dataset, GlobalOpinionQA, all the questions and answers are in English. The purpose of this dataset is to explore how well models respond to questions from different cultural backgrounds within an English context. When asking questions in English, we want the model to leverage feed-forward activations from non-English languages to better capture cultural nuances. Hence, for culture-aware tasks, we perform "non-En->En" XTransplant, where the questions are in English, but activations from non-English languages are transplanted into the English context. For example, when asking a question related to Chinese culture, we ask the question in English but feed-forward activations from Chinese are transplanted to help.
> >
> > I hope this clarifies the rationale behind the configurations we use for these distinct tasks.
> >
> >
> > > **5. Response to your question about "The 0.0 accuracy in the English subset of XCOPA for Qwen-2"**
> >
> > Thank you for pointing out the "0.0 accuracy" in the English subset of XCOPA for Qwen-2. We are sorry for not providing a detailed explanation of this result in our paper.
> >
> > Upon observing this result, we were also puzzled, so at that time, we specifically revisited Qwen-2's responses to the English subset of XCOPA.
> >
> > We found that the "0.0 accuracy" issue stemmed from the model's failure to effectively follow the instructions in our prompt. The exact prompt we used was:
> > ```
> > You are assigned to complete a two-category classification task.
> >
> > Premise: The girl squeezed her nose.
> > Options: (1) The baby drools on the bib.
> > (2) The baby soiled his diaper.
> >
> > Please determine which of the two options is more likely to be the cause of the given premise.
> >
> > Your Answer:
> > ```
> >
> > However, Qwen-2's responses were as follows:
> > ```
> > " Option 1 (The baby drools on the bib) is less likely to be the cause of"
> > " Option 1, \"The audience clapped their hands to the music,\" is more likely to be"
> > " Option 1 is more likely to be the result of the given premise. If the man expected the"
> > " Option 2, \"Her opponent felt sorry for her,\" is more likely to be the result of"
> > " Option 2, The products are made by child labor. \n\nExplanation: The premise states that radicals"
> > " Option 2, \"It's snack time,\" is more likely to be the cause of the given"
> > ...
> > ```
> >
> > Our evaluation script considers a model's response correct only if it contains the correct option (e.g., (1) or (2)) and excludes all other options. But as you can see above, Qwen-2's responses do not match this format, leading to the "0.0 accuracy".
> >
> > To ensure fairness in evaluation, we can not arbitrarily modify our evaluation script based solely on Qwen's responses on the English subset of the XCOPA dataset. Therefore, we have retained this result in our main experimental table.
> >
> > We hope this clarifies the issue, and again, we are sorry for not clarifying this issue in the original manuscript.

---

> > > ### Author Response · Authors · 2024-11-19
> > > **Kind Reminder**
> > >
> > > I hope this message finds you well. Thank you for your detailed review and for raising your concerns about our work. We have carefully reviewed your comments and provided responses to clarify the points raised.
> > >
> > > We greatly value your feedback and understand that certain aspects of our work might have led to some misunderstandings. If you have additional comments or would like to further discuss any points to ensure mutual understanding, we would be more than happy to engage in continued dialogue.
> > >
> > > Thank you once again for your time and consideration. We look forward to hearing from you.

---

> > > ### Author Response · Authors · 2024-11-20
> > > **New baseline of Multilingual Supervised Fine-tuning supplemented**
> > >
> > > We are excited to share additional baseline results for your consideration.
> > >
> > > We have included multilingual supervised fine-tuning in our experiment to further illustrate the extent to which multilingual capabilities can be unlocked through the XTransplant mechanism.
> > >
> > > Our results show that multilingual supervised fine-tuning indeed enhances the multilingual capabilities of LLMs. The comparison also reveals that the cross-lingual latent representation interaction enabled by XTransplant not only offers substantial benefits but also has a strong chance of surpassing the improvements achieved through additional supervised fine-tuning, demonstrating an innovative and highly promising direction for extending the boundaries of LLM performance.
> > >
> > > Detailed results and conclusions are in Table 1 and Section 4.2 in our newly revised and updated PDF file.
> > >
> > > We hope this additional message further clarifies the significance of our XTransplant mechanism.

---

> > > ### Comment · Reviewer_NGzZ · 2024-11-21
> > >
> > > Thank you for your response. I am still concerned about the first point, i.e., the "possibility of lurking into the gold answer." Following your instructions, I took a look at the code. In each evaluation file, we have the following code for evaluating UpperBound (an example comes from `upper_XNLI_reverse.py` lines 109 - 111):
> > >
> > > >sorted_list = sorted(flattened, key=lambda x: x[1], reverse=True)
> > > >
> > > >best_case = sorted_list[0]
> > > >
> > > >print(lang, best_case[0], best_case[1] / 50, "||| Upper Bound:", len(set(grid[N, N])) / 50)
> > >
> > >
> > > In each step, the best accuracy is selected for N x N transplatation options. For instance, in `Llama2-7B`, the answer of Upperbound is considered correct if the prediction is correct in ANY of 1024 transplantation options, as N=32 . In other words, in inference, UpperBound adds an expressive parameter (src $\in$ 0,.., 31; tgt $\in$ 0,..., 31), which values are set based on the correct answer.
> > > Therefore, I maintain my criticism that "UpperBound lurks into the gold answer" during inference.
> > >
> > > I do not see a parallel with Huang, Liu et al. 2018, who proposed a new architecture with information flow enabling "feature reuse." Their architecture was set before training and did not include greedy alternation during test time like UpperBound.
> > >
> > > I also appreciate the auhtors' answers to other points, yet they have not changed my opinion.

---

> ### Author Response · Authors · 2024-11-21
> **Clarifications and Responses to your Further Comments**
>
> We sincerely thank you for your response to our previous clarification. In particular, we deeply appreciate your diligence in carefully reviewing our code. However, **it seems there are still some misunderstanding regarding our work**.
>
> **Regarding your unresolved concern about the "golden answer leakage"**
> - In our previous reply, we mistakenly assumed that your concern about golden answer leakage was related to the Transplantation process in XTransplant, so we clarified that the Transplantation process does not involve any interaction with the gold answer. However, from your latest response, **we now realize that your concern is actually about *golden answer leakage* during the evaluation stage**. We apologize for this misunderstanding and appreciate the opportunity to clarify further.
>
> - We confirm that your interpretation of our evaluation code is correct, and we sincerely thank you for thoroughly examining it. That said, we would like to emphasize that the purpose of this *Upper Bound* evaluation is specifically to benchmark the potential upper limit of cross-lingual latent interaction achieved by XTransplant under our experimental setup.
>
> - While we understand that the high *Upper Bound* value may raise skepticism, we want to make it clear that **we are not using these results to claim the superiority of XTransplant over other baseline methods**. In fact, we **make no such assertions in the conclusions of our experiments**. Instead, the high *Upper Bound* results are intended **to illustrate the extent to which multilingual capabilities can be unlocked through the XTransplant mechanism without modifying LLM itself**. **It's totally different** from the common evaluation of directly comparing numerical results to demonstrate the superiority of one method over another.
>
> - Additionally, from your comments, we noticed your concern that achieving the *Upper Bound* from the N^2 (e.g., N^2=32^2=1024 for LLaMA-2) possible combinations  might be coincidental. However, as discussed in *Section 5.3: Analysis of Layer-Specific Effectiveness in Source and Target Selection*, one of our findings suggests that even when computing the *Upper Bound* from a space of size N, XTransplant can still achieve a satisfactory upper bound performance.
>
> - Moreover, another part of our paper—**Figure 7** in Section "Proportion Analysis of XTransplant Outcomes"—further supports that the high *Upper Bound* is not coincidental. In Figure 7, the filled area represent the proportion of all N^2 outcomes where XTransplant produces correct answers, while the shaded area represent incorrect answers. We observe that the proportion of correct answers is significantly high across all datasets, often exceeding 50%.
>
> **Regarding your concern about the "our explanation to the improvements under English2English setting"**
>
> - We would like to clarify the underlying principles behind the improvements observed with XTransplant in the English2English setting, which, to some extent, are indeed similar to the principles of "DenseNet" (Huang, Liu et al. 2018). Reviewer 4 has acknowledged this connection, and we are happy to provide a more detailed explanation for your understanding.
> - In non-En->En or En->non-En scenarios, XTransplant performs latent interactions between two different language versions of the same input. However, in the En->En setting, XTransplant simplifies to replacing the feed-forward activations between different decoder layers within the **same input** (for example, the feed-forward activations of the 15th layer can be replaced with those from the 25th layer, and the forward propagation can then continue from the 15th layer with the newly transplanted feed-forward activations). **This operation can be thought of as establishing interaction connections between different decoder layers, allowing feature (or knowledge) sharing between these layers to some extent. This is why we describe it as strengthening feature propagation and encouraging feature reuse**. This form of cross-layer connection is indeed quite similar to the connections in "DenseNet", which we believe provides a reasonable explanation for the improvements observed in the English2English setting. And of course, as you correctly pointed out, the concept of Upper Bound is not related to "DenseNet".
>
> We hope these clarifications address your concerns and provide further confidence in our work.

---

> > ### Comment · Reviewer_NGzZ · 2024-11-21
> >
> > Thank you for providing further clarification.
> >
> > I noticed that in the updated version, you have added a sentence clarifying that the aim of your experiments is not to claim improvement over standard LLM predictions. This edition helps understanding your motivations.
> >
> > Nevertheless, I still think that the paper, in its current form, does not present evidence to what extent cross-lingual transplantation improves multilingual capabilities. As noted in your explanations to en->en results, significant improvements can be achieved mainly by "feature reusing." More constrained experimental settings (e.g., transplantation from ith to ith layer across languages) would be needed to highlight the role of cross-language feature transfer while marginalizing "cross-layer" feature transfer.
> >
> > I see great potential in your work. However, after considering the points above and the discussion, I decided to keep my score for this submission. Thank you for understanding!

---

> > > ### Author Response · Authors · 2024-11-22
> > > **Thank you for your reply**
> > >
> > > Thank you for your reply. Though our explanation did not ultimately change your decision, we fully respect it. We sincerely appreciate the time and effort you have dedicated during the review and discussion phases.

---

### Official Review · Reviewer_a6n1 · 2024-11-04

**Soundness:** 2
**Presentation:** 3
**Contribution:** 2
**Rating:** 5
**Confidence:** 3

**Summary:**

In this paper, the authors propose a new method, XTransplant, to exploit English-based capabilities within non-English contexts using English-centric Large Language Models (LLMs). For this purpose, XTransplant utilizes the feed-forward activations of a decoder layer on English text into a layer of the decoder on non-English input (or from non-English into English) while predicting the first new token. To select optimal pairs of layers in different languages, the authors investigate all possible combinations and their performance, referred to as the instance-aware upper bound in this paper. The experimental results on XNLI, XQuAD, and XCOPA with LLaMA-2-7B-Chat, Mistral-7B-Instruct-v0.3, Qwen2-7B-Instruct, and Chinese-Alpaca-2-7B show that the proposed XTransplant improve the performance under the setting of the instance-aware upper bound.

**Strengths:**

- In an ideal situation, XTransplant can enhance task-solving performance by accessing the knowledge of centric languages like English.
- In an ideal situation, XTransplant can work on both English- and Chinese-centric LLMs.
- This work investigates all possible pairs of source and target layers.

**Weaknesses:**

- In the experiments, the authors compared the upper bound results of XTransplant using the best combination of layers with baseline results. This is unfair and should not be reported as the main result of XTransplant. Reporting the result for analyses is acceptable. Instead, the authors can use the average or median performance in Figure 5 as the main result.
- For fair comparison, the authors need to decide the pair of source and target layers based on the performance of validation data.
- Furthermore, the test set size for each language is limited to 50 instances. This size is quite small. The authors need to consider the variance of the results for each language in such a small setting. Thus, increasing the test set size is required to make the results more reliable.
- When targeting cross-lingual tasks, utilizing machine translation is one of the easiest way. However, such a basic approach is not considered as a baseline in the paper. Instead, the authors concatenate the two different languages in PIM, a baseline approach.
- Considering the computational inefficiency of choosing the upper bound pairs of layers, reporting the computational cost of doing that is also required.

**Questions:**

- What is the reason for applying XTransplant only when generating the first new token?

---

> ### Author Response · Authors · 2024-11-13
> **Reply to your comments (part 1)**
>
> > **1. Response to your concern about "The unfairness of comparisons"**
>
> Thank you for your feedback. It seems there might be a **misunderstanding** regarding the purpose of our main experiments. While we have presented both the baseline results and the upper bound results of XTransplant in the same table, our intention was not to claim that XTransplant is a superior method to the other baselines. We do not make any such comparison or assertion in the conclusions of our experiment.
>
> In fact, one of the key objectives of our main experiments is to highlight that both the PIM results and the upper bound results of XTransplant show performance improvements beyond the original model’s capabilities. This serves to demonstrate that, for current LLMs, simply prompting them under multilingual or culture-aware contexts does not fully exploit their multilingual potential. And by showcasing the upper bound results of XTransplant, we aim to highlight that there is still significant room for improvement.
>
> Additionally, we would like to emphasize that **XTransplant is not a specific methodology but rather a mechanism** — one that has not been explored in previous works — using cross-lingual feed-forward activation transplantation.
>
> We hope this clarifies our intentions.
>
> > **2. Response to your concern about "The test set is too small"**
>
> As you pointed out, we have retained 50 instances for each language in the test set (aka the pilotset in our paper). The specific details are as follows:
>
> **「Involved Languages / Cultures」**
>
> XNLI (15): ar, bg, de, el, en, es, fr, hi, ru, sw, th, tr, ur, vi, zh
>
> XQuAD (12): ar, de, el, en, es, hi, ro, ru, th, tr, vi, zh
>
> XCOPA (11): en, et, ht, id, it, sw, ta, th, tr, vi, zh
>
> GlobalOpinionQA (24 cultures): am, ar, bn, de, el, en, es, fr, hi, id, it, ja, ko, nl, pt, ru, sv, sw, tl, tr, uk, ur, vi, zh-CN (QA pairs are all presented in English)
>
> **「Sample Size (50 samples per language / culture)」**
>
> XNLI: $50 \times 15 = 750$
>
> XQuAD: $50 \times 12 = 600$
>
> XCOPA: $50 \times 11 = 550$
>
> GlobalOpinionQA: $50 \times 24 = 1200$
>
> And for multilingual tasks (XNLI, XQuAD, XCOPA), we ensure that each language set consists of the same 50 questions, presented in different language versions.
>
> The reason we have sampled only 50 instances per language is due to the extensive scale of our experiments. As noted between "lines 213–215" of our manuscript, to obtain the instance-aware upper bound of XTransplant, we perform inference on all N^2 possible source and target layer selection strategies for each instance. For example, in LLaMA-2-7B-Chat with N=32, N^2=1024 times inferences are conducted per instance. Our main experiments involve 3 LLMs and 4 pilotset datasets, resulting in over 800 hours of computation on 8 * A800-SXM4-80GB.
>
> We appreciate you for pointing this out and hope you could understand the computational constraints.

---

> ### Author Response · Authors · 2024-11-13
> **Reply to your comments (part 2)**
>
> > **3. Response to your concern about "Not utilizing machine translation as a baseline"**
>
> Thank you for pointing this out. However, we would like to clarify that the "translate-then-xxx" pipeline is not suitable for the tasks in our study. It seems there might be some aspects of our work that were not fully clear, which led to your concern about this.
>
> Our study involves 4 specific datasets: XNLI, XQuAD, XCOPA, and GlobalOpinionQA.
>
> - For the multilingual datasets (XNLI, XQuAD, and XCOPA), all of the questions are linguistically parallel across languages. This means that the questions in each language split are essentially the same, with differences only in the language versions. Therefore, the "English Set" in these datasets already represents the translated version, as you described with a "machine translation" approach. As such, there is no need to apply additional translation, which is why we do not consider "translate-then-xxx" as a suitable baseline for these tasks.
>
> - Regarding the multicultural dataset, GlobalOpinionQA, all the questions and answers are in English. The purpose of this dataset is to explore how well models respond to questions from different cultural backgrounds within an English context. Therefore, machine translation would not be applicable here either.
>
> We hope this clarifies the reason why we do not utilize "machine translation" as a baseline.
>
>
> > **4. Response to your concern about "The computational cost of choosing the upper bound pairs of layers"**
>
> - As emphasized in our first response, **XTransplant is not a specific methodology but rather a mechanism**. Therefore, our work does not focus on "how to choose the upper bound pairs of layers?" The essence of our work is to explore cross-lingual feed-forward activation transplantation as a bold, novel attempt in the area of enhancing multilingual capabilities and cross-lingual transfer. We begin by conducting extensive experiments (over 800 hours of computation on 8 * A800-SXM4-80GB) to demonstrate the underutilization of current LLMs' multilingual potential. And through the upper bound results of XTransplant and subsequent analysis, we try to show that XTransplant is a feasible and promising mechanism for improving both multilingual capabilities and cultural adaptability. While we certainly hope our work can inspire future research, such as the more efficient and accurate identification of appropriate pairs of layers, our focus is on the exploration and analysis of this novel cross-lingual feed-forward activation transplantation approach, rather than providing a specific methodology.
>
> - And it is important to emphasize that the extensive experiments serve to demonstrate the potential of XTransplant as a simple yet promising mechanism for enhancing multilingual capabilities and cross-lingual transfer, without any modifications to the LLM itself. **This does not imply that future work must replicate the same approach in terms of computational expense**.
>
> - As the first work to explore this mechanism, we felt that the effort spent on a comprehensive analysis was justified, and we believe it has successfully showcased the substantial potential of XTransplant.
>
> - Additionally, while our work involves significant computational expense, we do not intend for future research to replicate this approach in a resource-intensive manner. **Our goal is to inspire future work that can utilize the concept/mechanism of cross-lingual feed-forward activations transplantation to model design or training phases**, such as exploring cross-lingual feed-forward interactions and connections between different layers or language-specific regions, to enable the model better leverage its intrinsic multilingual knowledge.
>
> - Lastly, we hope that the large-scale experiments we conducted, along with our subsequent analysis and discussions, can inspire future research and provide valuable insights for the broader research community.
>
> > **5. Response to your question about "What is the reason for applying XTransplant only when generating the first new token?"**
>
> Thank you very much for noticing this question. We are sorry for not providing a more detailed explanation of this aspect in our paper.
>
> In autoregressive generation, applying XTransplant during the generation of the first new token essentially introduces the benefit of feed-forward activations from another language across the entire sequence generation process. This is because all subsequent tokens are influenced by the activations cached from earlier steps. If XTransplant were applied during the generation of every token, it would be a redundant operation and could even cause the model's output to break down.
>
> We hope this clarifies the reasoning behind.

---

> > ### Author Response · Authors · 2024-11-19
> > **Kind Reminder**
> >
> > I hope this message finds you well. Thank you for your detailed review and for raising your concerns about our work. We have carefully reviewed your comments and provided responses to clarify the points raised.
> >
> > We greatly value your feedback and understand that certain aspects of our work might have led to some misunderstandings. If you have additional comments or would like to further discuss any points to ensure mutual understanding, we would be more than happy to engage in continued dialogue.
> >
> > Thank you once again for your time and consideration. We look forward to hearing from you.

---

> > ### Author Response · Authors · 2024-11-20
> > **New baseline of Multilingual Supervised Fine-tuning supplemented**
> >
> > We are excited to share additional baseline results for your consideration.
> >
> > While "machine translation" may not be an ideal baseline for our work as we have explained, we have included multilingual supervised fine-tuning in our experiment to further illustrate the extent to which multilingual capabilities can be unlocked through the XTransplant mechanism.
> >
> > Our results show that multilingual supervised fine-tuning indeed enhances the multilingual capabilities of LLMs. The  comparison also reveals that the cross-lingual latent representation interaction enabled by XTransplant not only offers substantial benefits but also has a strong chance of surpassing the improvements achieved through additional supervised fine-tuning, demonstrating an innovative and highly promising direction for extending the boundaries of LLM performance.
> >
> > Detailed results and conclusions are in Table 1 and Section 4.2 in our newly revised and updated PDF file.
> >
> > We hope this additional message further clarifies the significance of our XTransplant mechanism.

---

> > > ### Comment · Reviewer_a6n1 · 2024-11-25
> > > **I've decided to increase my score**
> > >
> > > Thank you for your response.
> > >
> > > > This serves to demonstrate that, for current LLMs, simply prompting them under multilingual or culture-aware contexts does not fully exploit their multilingual potential. And by showcasing the upper bound results of XTransplant, we aim to highlight that there is still significant room for improvement.
> > >
> > > First of all, your approach is based on the test set tuning. If other prompting methods can do that, we can expect further gains. At least, I agree with your claim that "simply prompting them under multilingual or culture-aware contexts does not fully exploit their multilingual potential". However, the most important thing is how to solve this problem. Therefore, the contribution of this part is subtle in this paper.
> > >
> > > >The reason we have sampled only 50 instances per language is due to the extensive scale of our experiments. As noted between "lines 213–215" of our manuscript, to obtain the instance-aware upper bound of XTransplant, we perform inference on all N^2 possible source and target layer selection strategies for each instance. For example, in LLaMA-2-7B-Chat with N=32, N^2=1024 times inferences are conducted per instance. Our main experiments involve 3 LLMs and 4 pilotset datasets, resulting in over 800 hours of computation on 8 * A800-SXM4-80GB.
> > >
> > > I understand your situation by reading your response. However, it does not guarantee the stability of the evaluation results.
> > >
> > > >Thank you for pointing this out. However, we would like to clarify that the "translate-then-xxx" pipeline is not suitable for the tasks in our study. It seems there might be some aspects of our work that were not fully clear, which led to your concern about this.
> > >
> > > Thank you for the explanation! I understand the details of the dataset.
> > >
> > > >Our goal is to inspire future work that can utilize the concept/mechanism of cross-lingual feed-forward activations transplantation to model design or training phases
> > >
> > > I think this is a contribution even though the importance is less important than proposing a solution.
> > >
> > > >In autoregressive generation, applying XTransplant during the generation of the first new token essentially introduces the benefit of feed-forward activations from another language across the entire sequence generation process. This is because all subsequent tokens are influenced by the activations cached from earlier steps. If XTransplant were applied during the generation of every token, it would be a redundant operation and could even cause the model's output to break down.
> > >
> > > You can shortly explain the concept by using a keyword, attention sink token.
> > > FYI: https://arxiv.org/abs/2309.17453
> > >
> > > >Our results show that multilingual supervised fine-tuning indeed enhances the multilingual capabilities of LLMs. The comparison also reveals that the cross-lingual latent representation interaction enabled by XTransplant not only offers substantial benefits but also has a strong chance of surpassing the improvements achieved through additional supervised fine-tuning, demonstrating an innovative and highly promising direction for extending the boundaries of LLM performance.
> > >
> > > I appreciate your efforts. The newly added SFT results can deepen the discussion and raise your contribution. Considering the above response and the paper update, I've decided to increase my score.

---

> > > > ### Author Response · Authors · 2024-11-26
> > > > **Thank you very much for your valuable feedback and for increasing the score !**
> > > >
> > > > Thank you very much for your valuable feedback and for increasing the score. We greatly appreciate your thoughtful comments !

---

### Author Response · Authors · 2024-11-20
**General Message to All Reviewers**

Based on the valuable feedback from the reviewers, we have **revised our manuscript** accordingly and **updated the PDF file**. Below are some of the specific changes we have made:


1. **Line 53**: We have condensed the two sentences about "curse of multilinguality" and "negative interference", clarifying that the "curse of multilinguality" refers to a specific form of negative interference that arises when dealing with a large number of languages.

2. **Line 68-71**: We have **enriched the caption of Figure 1**, providing a more detailed description of how XTransplant works.


3. **Line 117-120**: In Section 2, we have elaborated on the rationale and motivations behind **our focus on "feed-forward activations"**.

```
The background, that the feed-forward layers have been shown in many studies to play a crucial role in storing factual knowledge[1][2][3], provides the reason for why we explore cross-lingual transplantation on feed-forward layers, which aligns with our goal of enabling LLMs to fully leverage both English and non-English multilingual knowledge.

[1] Transformer feed-forward layers are key-value memories.
[2] Knowledge neurons in pretrained transformers.
[3] Locating and editing factual associations in gpt.
```

4. **Line 162**: We have changed the title of Section 3.2 from "Bi-directional Transplantation" to **"Mutual Transplantation"** to better reflect the nature of XTransplant. Additionally, some statements in this section have also been refined to more clearly articulate the mutual transplantation process.
5. **Line 198-207**: In Section 4.1, we have revised the introduction to our involved datasets  to provide a clearer explanation of **why we use the En->non-En configuration in multilingual tasks and the non-En->En configuration in culture-aware tasks**.
6. **Line 264-268**: In Section 4.2, we further emphasize the purpose of our main experiment.
   - While we present both the baseline results and the upper bound results of XTransplant in the same table or figure, our goal is not to demonstrate the superiority of XTransplant. Rather, we aim to use these comparisons **to illustrate the extent to which multilingual capabilities can be unlocked through the XTransplant mechanism without modifying the LLM itself**.
7. **Line 259-260**: In our main experiment, we introduced **Multilingual SFT as a new baseline**, using over 20k multilingual instruction data for further supervised fine-tuning of LLMs to enhance their multilingual capabilities. And we have the following observation (**Line 284-288**):
   - The comparison with Multilingual SFT shows that the cross-lingual latent representation interaction enabled by XTransplant mechanism not only offers substantial benefits but also has a strong chance of surpassing the improvements achieved through additional supervised fine-tuning, demonstrating an innovative and highly promising direction for extending the boundaries of LLM performance.
8. **Line 248**: We have added a footnote in the English subset of XCOPA for Qwen2-7B-Instruct in Table 1, directing readers to Appendix B.5 for a detailed explanation.
9. **Line 289-297**: we provide additional clarification on the phenomenon of **"Improvements under the English2English setting"** and explain its rationale.
10. **Line 323**: "Proportion Analysis of XTransplant Outcomes" is moved from main body to Appendix C.1
11. **Line 326-327**: We have provided additional clarification on the **definition of "language consistency"** to enhance the reader's understanding.
12. **Line 810**: We have added a section titled "Potential Questions and Explanations" in the Appendix to further emphasize certain aspects of our work and provide explanations for potential questions about our work.

---

### Note · Authors · 2024-12-11

**Comment:**

We sincerely thank all the reviewers for their valuable feedback. Through their comments, we have recognized the shortcomings in our work and are committed to improving it in future versions. As a result, we have decided to withdrawal the current submission.

**Withdrawal Confirmation:**

I have read and agree with the venue's withdrawal policy on behalf of myself and my co-authors.